# Can LLM "Self-report"?: Evaluating the Validity of Self-report Scales in Measuring Personality Design in LLM-based Chatbots

**Huiqi Zou[1], Pengda Wang[2], Zihan Yan[3], Tianjun Sun[2], and Ziang Xiao[1]**

[1]Department of Computer Science, Johns Hopkins University
[2]Department of Psychological Sciences, Rice University
[3]Cognitive Science and Language Processing Informatics,
 University of Illinois Urbana-Champaign
{hzou11,ziang.xiao}@jhu.edu; {pw32,ts110}@rice.edu; zihan25@illinois.edu

## Abstract

A chatbot's personality design is key to interaction quality. As chatbots evolved from rule-based systems to those powered by large language models (LLMs), evaluating the effectiveness of their personality design has become increasingly complex, particularly due to the open-ended nature of interactions. A recent and widely adopted method for assessing the personality design of LLM-based chatbots is the use of self-report questionnaires. These questionnaires, often borrowed from established human personality inventories, ask the chatbot to rate itself on various personality traits. Can LLM-based chatbots meaningfully "self-report" their personality? We created 500 chatbots with distinct personality designs and evaluated the validity of their self-report personality scores by examining human perceptions formed during interactions with these chatbots. Our findings indicate that the chatbot's answers on human personality scales exhibit weak correlations with both human-perceived personality traits and the overall interaction quality. These findings raise concerns about both the criterion validity and the predictive validity of self-report methods in this context. Further analysis revealed the role of task context and interaction in the chatbot's personality design assessment. We further discuss design implications for creating more contextualized and interactive evaluation.

## 1 Introduction

With recent advancements in artificial intelligence (AI), particularly in natural language processing (NLP), AI-based chatbots have found applications across various fields. Their use in areas, such as creative writing (Chung et al., 2022), mental health counseling (Lai et al., 2023), data collection (Wei et al., 2024) and educational support as teaching assistants (Hicke et al., 2023), has made interactions with conversational agents increasingly common. To further enhance dialogue engagement and tailor these agents to specific tasks, developers often manipulate chatbots' behaviors or assign them particular identities. One prevalent strategy is through personality design (Shao et al., 2023; Li et al., 2023; Park et al., 2023).

The design of chatbot personalities has evolved significantly, evolving from early rule-based systems such as *ELIZA* (Weizenbaum, 1966), which relied on predefined scripts to simulate human conversation, to contemporary approaches based on large language models (LLMs). The involvement of LLMs enables various methods of personality instantiation, such as characterizing agents through demographic and persona-based dialogue, as well as employing more fine-grained approaches. For instance, Goldberg (1992)'s set of 70 bipolar adjectives, mapped to the Big Five personality domains (Serapio-García et al., 2023), and their facets have been used to characterize chatbot behaviors. Similarly, researchers have transformed personality scale items into open-ended prompts to elicit personality-consistent responses from LLMs (Ran et al., 2024).

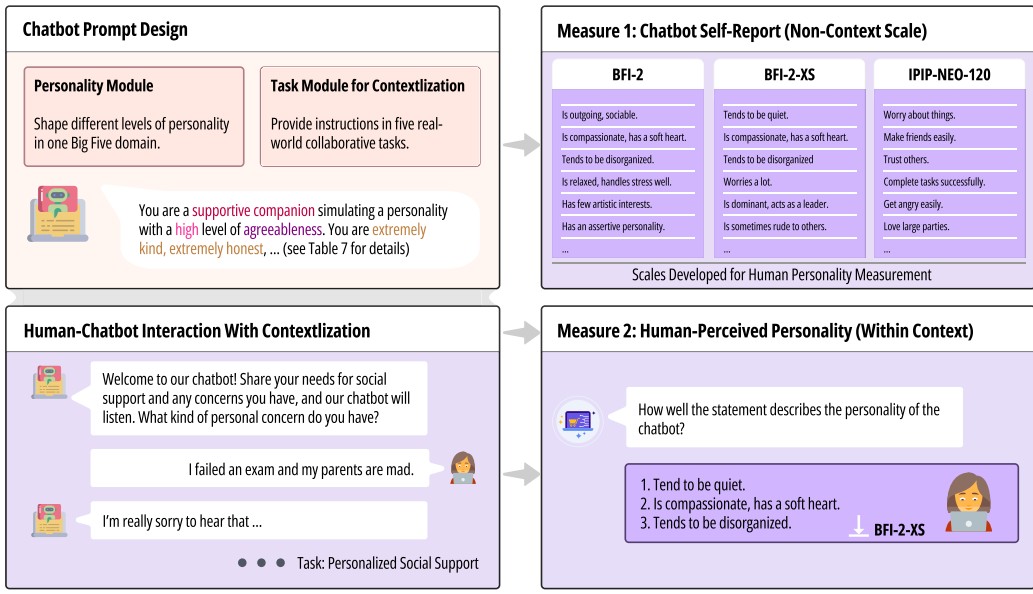

Figure 1: Overview of the Evaluation Pipeline. Two measures are used to evaluate the chatbot's personality: (1) Chabot self-reporting through established personality scales (e.g., BFI-2 and IPIP-NEO-120); (2) Human-perceived personality, where human interact with the chatbot and subsequently rate its personality based on their interaction.

As chatbots become increasingly open-ended in their capabilities, evaluating the effectiveness of their personality design presents growing challenges. Unlike earlier rule-based systems, modern chatbot architectures lack hand-crafted rules that designers can easily examine or adjust. However, this openness also facilitates the development of novel evaluation methods. Recent research has begun to explore the use of self-report personality scales, originally developed for assessing human traits, to evaluate the personality constructs of LLM-based chatbots (Huang et al., 2023a; Wang et al., 2024c; Tu et al., 2023). In these studies, LLMs are typically prompted to respond to items from standardized personality inventories using a Likert scale format (Likert, 1932), where each item works as an indicator to reflect a specific personality construct dimension. Responses to individual items are summarized to compute aggregate scores for each personality domain.

Although the "self-report" method is convenient and cost-effective for evaluating LLMs, it relies on assumptions that have not yet been systematically validated through empirical research. First, it assumes a high degree of alignment between the personality traits attributed to an LLM-based chatbot by design and those perceived by humans during interactions (the ultimate goal of evaluation). However, even in human psychological research, the consistency between self-reports and informant reports tends to be limited due to differences in informational sources (e.g., Connolly et al., 2007; Connelly & Ones, 2010; McCrae & Costa, 1987). Second, these methods presume that the structure and content of psychological assessment tools developed for humans can also be applied to LLM-based chatbots. For instance, a chatbot designed for project management might be assessed on Extraversion using an item like "Love surprise parties" (Johnson, 2014a), which is irrelevant to the chatbot's actual role in the designed task. Although these psychometric scales are well-validated for human samples, the examined validity does not directly transfer to chatbot evaluation, raising concerns about the applicability of such methods in non-human settings.

This study aims to fill the gap by evaluating the validity of an LLM-based chatbot's self-report personality. Guided by principles from classical test theory and measurement theory (e.g., Allen & Yen, 2001; Xiao et al., 2023), we evaluated two sets of validity: Convergent and Discriminant validity, where we examined the intrinsic structure, and Criterion and Predictive validity, where we examined the extrinsic relationship with relevant variables.

In particular, we focus on two external variables central to the ultimate goal of personality design in AI systems: human perception and interaction quality.

In this study, we created 500 chatbots, each with distinct personality designs, and collected their "self-report" personality. We recruited 500 human participants, each assigned to interact with one of the chatbots to complete a designated task, evaluate the chatbot's personality, and rate the interaction quality. Figure 1 presents an overview of the evaluation pipeline. Our findings indicate that while the self-report personality scales demonstrate moderate Convergent and Discriminant validity, they show poor alignment with human-perceived personality scores and weak correlations with the human-rated interaction quality, suggesting limited Criterion and Predictive validity.

Our study reveals the limitations of relying solely on self-report measures to assess personality design in LLM-based chatbots. We argue that effective evaluation methods must account for how a chatbot's personality emerges through interaction with human users and should be grounded in task-based performance. Future research should consider the rich signals in human-chatbot interactions and incorporate human perception as a critical dimension in evaluating personalized AI systems. Our work makes three main contributions:

- Through an empirical study with 500 participants, we highlight the validity concerns of using self-report personality scales to evaluate the personality design of LLM-based chatbots.

- Our result offers design implications for creating effective personality design evaluation methods that are grounded in real-world task interactions.

- We present a dataset containing a rich log of human interactions with 500 chatbots, each with distinct personality designs and human perceptions of their personalities, facilitating the development of interaction-based personality evaluation methods.

## 2 Related Work

**Human-Like Responses in LLMs.** Recent studies have shown that LLMs demonstrate remarkable abilities resembling human-like responses. For example, Lampinen et al. (2024) found that LLMs display response patterns similar to humans in logical reasoning tasks, with similar findings being reported by Kosinski (2023) and Wang et al. (2024a). Additionally, Pellert et al. (2023) found that LLMs perform comparably to humans across various psychological scales, such as value orientations and moral norms. All these abilities have garnered significant interest from both computer scientists and social scientists, as these findings suggest that we may be able to draw upon existing social science research on humans as a reference for LLM studies. For instance, personality, which has been extensively studied in psychology and consistently proven to be a reliable predictor of various behavioral and psychological outcomes, is one such area of interest. Personality encodes rich and complex information in textual data (Goldberg, 1990; Saucier & Goldberg, 2001). LLMs may capture and model such encoding by learning from vast training data. Many studies explore the personality traits LLMs exhibit, with the hope to leverage these simulated traits for predicting or shaping model behavior (Lee et al., 2024; Huang et al., 2023b; Wang et al., 2024b; 2025; Ran et al., 2024).

**Current Approaches for Measuring LLM-based Chatbots' Personality.** Currently, most assessments of LLM-based chatbot personality rely on psychological frameworks developed for human personality measurement, typically employing standardized personality scales. A common approach is to have the model respond to scales that have already been developed and validated on human samples, such as the Big Five Inventory-2 (BFI-2) (Soto & John, 2017a) and HEXACO (Lee & Ashton, 2004; 2006). The model's scores are then calculated based on its answers to the multiple-choice questions (MCQs) on these scales. The rationale behind this is straightforward: since we believe that LLMs simulate human abilities, we can directly borrow well-established and validated methods used for humans to assess the personality traits of these models. For example, studies like Huang et al. (2023b) used this approach to evaluate whether LLMs reflect the personality traits outlined in the BFI (John

et al., 1999). Serapio-García et al. (2023) presented the reliability and validity of LLMs in resembling personality profiles along desired dimensions with International Personality Item Pool– Neuroticism, Extraversion, Openness (IPIP-NEO) (Goldberg et al., 1999) and BFI (John et al., 1999). However, some researchers argue that traditional scales may not sufficiently capture the personality of LLMs and have adopted alternative approaches. For instance, some researchers modified scale questions into prompts to elicit text-based responses (Wang et al., 2024c), while others have experts rate stories generated by LLMs according to specified criteria (Jiang et al., 2023). These evaluation methods may more accurately reflect real-world application scenarios.

**Self-Reports vs. Informant Reports.** Self-report methods have limitations in capturing the behavior nuances that reflect personality traits of LLM-based chatbots, as they are not grounded in user experience. Given that the primary goal of chatbot personality design is to enhance engagement and provide personalized interactions, assessment methods should center on human perception. This challenge parallels long-standing exploration in psychology regarding the distinction between self-reports and informant reports (e.g., Connolly et al., 2007; Connelly & Ones, 2010; McCrae & Costa, 1987). While self-reports capture an individual's subjective understanding of their own traits, informant reports offer an external perspective based on others' interpretations of the individual's behavior. A meta-analysis in psychology found that the average correlation between self-reports and informant reports for the Big Five personality traits was only 0.36 (Connolly et al., 2007), indicating overlap and divergence. For chatbots, task-specific design further influences personality expression, potentially widening the gap between their self-report and human-perceived traits. Thus, relying solely on self-reports may fail to reflect how chatbots actually affect user experience.

## 3 Methodology

### 3.1 Chatbot Design

In this study, we first created a set of LLM-based chatbots with various personality designs. Those chatbots were created for tasks in which personality design is critical for user experience. Our chatbot framework consists of two main components: a personality module (§3.1.1) and a task module (§3.1.2). The personality module shapes the chatbot's behavior with predefined personality traits, while the task module provides role-specific instructions and task outlines to complete the task. The complete prompts are in Appendix A.

#### 3.1.1 Personality Module

The personality design of our chatbot is based on the Big Five model (John et al., 1999), a foundational framework in personality research that describes personality across five key domains: extraversion (EXT), agreeableness (AGR), conscientiousness (CON), neuroticism (NEU), and openness to experience (OPE). We adopted the *shape* approach from Serapio-García et al. (2023) to shape a chatbot's personality in each domain. This method extends Goldberg (1992) list of 70 bipolar adjectives to 104 personality trait descriptors, which are mapped to different facets within each domain. For example, adjectives such as 'unenergetic' and 'energetic' represent the lower and higher levels of Extraversion, respectively. These adjectives were then paired with linguistic qualifiers from Likert-type scales (Likert, 1932) to create personality profiles at varying levels. For our study, we selected the strongest linguistic qualifier (i.e., 'extremely') and randomly sampled five high-level or low-level adjectives from the same personality domain to generate prompts. Each chatbot is prompted to exhibit either a high or low level of a single personality trait, with others left unconstrained.

#### 3.1.2 Task Module

We selected five chatbot task categories where personality design plays a critical role in user experience and task effectiveness. This approach allows us to evaluate the validity of the chatbot's personality design via emulating real-world scenarios where designers rely on personality evaluation results to build an effective chatbot. The selected tasks are as follows:

1. **Job Interview**, where the chatbot works as an HR representative to assess the interviewee's Organizational Citizenship Behaviors (OCB), such as initiative, helping, and compliance, was chosen due to the strong association of the interviewer's conscientiousness in structured interview settings (Heimann et al., 2021).

2. **Public Service**, focusing on handling workday challenges like crisis management, highlights the chatbot's ability to help humans navigate high-stress situations where agreeableness and emotional stability are crucial (Mishra et al., 2023).

3. **Personalized Social Support**, where chatbot features are rooted in the counseling psychology principles of empathy and reflective listening as outlined by Rogers (1957), offers empathetic responses to individuals in distress referring Acceptance and Commitment Therapy (ACT) techniques. Research shows that empathy is closely linked to agreeableness (Walton et al., 2023).

4. **Customized Travel Planning**, which relies on the chatbot's decision-making and personalization skills, underscores the importance of anthropomorphic traits in recommendation systems that foster trust and enhance user decision-making (Qiu & Benbasat, 2009). Previous research also suggests that question types related to extraversion levels significantly impact user satisfaction (Miyama & Okada, 2022).

5. **Guided Learning**, where the chatbot aids humans in understanding complex topics, is based on the constructivist learning approach (Higgs et al., 2004) and highlights the critical role of extraversion in instructors for enhancing student acceptance.

In total, we randomly sampled five personality adjective markers at high or low levels of each domain ten times, respectively. Combined with five task instructions, this resulted in 500 distinct chatbot personality designs. We used GPT-4o (OpenAI, 2024), one of the most performant models available, as the backbone model with the temperature set to zero.

## 3.2 Personality Design Evaluation

### 3.2.1 Self-report Personality

To quantify the chatbot's self-report personality, we utilized three well-established psychometric tests that assess the Big Five domains (John et al., 1999): Big Five Inventory–2 Extra-Short Form (BFI-2-XS) (Soto & John, 2017b), BFI-2 (Soto & John, 2017a) and IPIP-NEO-120 (Johnson, 2014b). The chatbot's responses to test items were aggregated to calculate the personality scores for the five domains respectively, which were then used for subsequent analysis. Following the work of Serapio-García et al. (2023), we instructed the chatbot to rate test items using a standardized response scale from one to five based on the personality descriptions. The complete prompt format is shown in Appendix B.

### 3.2.2 Human-perceived Personality

We conducted a human study to gather participants' perceptions of the chatbot's personality with the designed interface (Appendix C). Each participant was randomly assigned to interact with one chatbot and engaged in multi-turn, task-based conversations.

After completing the task, participants were asked to rate their perceived personality of the chatbot using the BFI-2-XS (Soto & John, 2017b). The BFI-2-XS consists of 15 items, each representing a distinct facet of one of the Big Five personality domains, thus preserving the scale's comprehensive descriptive and predictive capabilities. To mitigate the impact of individual variability, we aggregated evaluation scores across chatbots sharing the same personality configuration (i.e., for all high or all low) for analysis.

## 3.3 Study Procedure

To gather the human-perceived personality of each chatbot, we recruited English-speaking participants from Prolific[1], with each participant assessing one chatbot. At the beginning

---

[1]https://www.prolific.com

of the survey, participants were given the task's objective, along with clear instructions on how to start the conversation, including an example to clarify how the participant might approach the task. A privacy reminder is included to ensure no private information is shared. After interacting with the chatbot, each participant filled out the chatbot personality evaluation questionnaire and completed a separate User Experience Questionnaire (UEQ) (Laugwitz et al., 2008). The entire process was designed to take 10 to 15 minutes. Appendix E presents a summary of the collected dataset statistics. Appendix F presents detailed participant statistics and their demographic information.

### 3.4 Analysis Method

Our analysis evaluates the effectiveness of personality settings and the alignment between self-reports and human perceptions using descriptive statistics and correlation analyses. We began by computing the means and standard deviations of both self-report and human-perceived personality scores to examine overall distribution patterns. Statistical significance tests were conducted to evaluate differences between the high and low personality conditions, as well as between human perceptions and the chatbot's self-report scores. In addition, we assessed construct validity, the extent to which a test accurately measures what it is intended to measure (Cronbach & Meehl, 1955), based on four psychometric dimensions:

- **Convergent validity** assesses whether different methods measuring the same trait yield consistent results (John & Benet-Martínez, 2000). To assess this, we compared the chatbot's self-report personality scores across three established scales. Strong inter-scale correlations indicate consistent responses under our personality settings.

- **Discriminant validity** evaluates the distinctiveness of different traits (Campbell & Fiske, 1959). We calculated correlations between different personality trait scores across psychometric scales and evaluation methods. Low correlations between unrelated traits confirm good discriminant validity.

- **Criterion validity** examines the correlation between a measure and an external criterion (Cronbach & Meehl, 1955). We assessed this by comparing the chatbot's self-report personality traits with human perceptions, measured through participant ratings after interacting with the chatbot. The value of criterion validity indicates the level of alignment between self-reports and human perceptions.

- **Predictive validity** refers to a measure's ability to predict future outcomes or behaviors (Funder, 2006). We evaluated this by examining whether the chatbot personality setting predicted the quality of interaction, as captured through the UEQ. Strong correlations between personality evaluation scores and positive user experiences suggest that personality traits are valid predictors of chatbot effectiveness.

To assess convergent and discriminant validity, we employed the Multitrait-Multimethod (MTMM) matrix approach (Campbell & Fiske, 1959), which evaluates multiple traits across different methods to identify patterns of consistency and distinctiveness. We conducted this analysis using Spearman correlations. Given the non-hierarchical structure of the BFI-2-XS, our analysis focused on domain-level personality traits.

## 4 Results

### 4.1 Personality Settings Work for Both Human and Chatbot

**Personality setting work but chatbot's self-report scores vary more significantly across personality conditions.** To evaluate the effectiveness of our personality settings, we examined descriptive statistics of self-report and human-perceived scores. Specifically, we tested for significant between-group differences under high and low personality conditions, which indicates successful trait manipulation. As shown in Table 1, self-report scores are consistently higher in high-setting conditions across tasks. Human-perceived scores followed a similar trend, except for Conscientiousness in the social support task. Traits that are more readily conveyed through language and interaction style (Mairesse & Walker, 2010),

| Evaluation | Domain | Job Interview | | | | Public Service | | | | Social Support | | | | Travel Planning | | | | Guided Learning | | | |
|---|---|---|---|---|---|---|---|---|---|---|---|---|---|---|---|---|---|---|---|---|---|
| | | High | | Low | | High | | Low | | High | | Low | | High | | Low | | High | | Low | |
| | | M | SD | M | SD | M | SD | M | SD | M | SD | M | SD | M | SD | M | SD | M | SD | M | SD |
| Self-report | EXT | 4.86 | 0.16 | 1.32 | 0.14 | 4.82 | 0.19 | 1.19 | 0.15 | 4.91 | 0.11 | 1.27 | 0.09 | 4.94 | 0.11 | 1.01 | 0.04 | 4.79 | 0.21 | 1.29 | 0.10 |
| | AGR | 4.56 | 0.10 | 1.75 | 0.13 | 4.98 | 0.02 | 1.39 | 0.24 | 4.99 | 0.03 | 1.70 | 0.28 | 4.95 | 0.07 | 1.01 | 0.02 | 4.86 | 0.11 | 1.63 | 0.31 |
| | CON | 4.68 | 0.13 | 1.44 | 0.26 | 4.99 | 0.02 | 1.06 | 0.06 | 4.47 | 0.23 | 1.10 | 0.08 | 5.00 | 0.00 | 1.02 | 0.03 | 4.96 | 0.07 | 1.09 | 0.08 |
| | NEU | 4.85 | 0.16 | 1.06 | 0.06 | 4.92 | 0.05 | 1.02 | 0.02 | 4.87 | 0.08 | 1.17 | 0.17 | 4.96 | 0.04 | 1.04 | 0.01 | 4.74 | 0.18 | 1.03 | 0.02 |
| | OPE | 4.83 | 0.18 | 1.45 | 0.11 | 4.98 | 0.02 | 1.32 | 0.09 | 4.75 | 0.19 | 1.31 | 0.19 | 4.97 | 0.06 | 1.03 | 0.06 | 4.88 | 0.31 | 1.52 | 0.12 |
| Human | EXT | 3.70 | 0.74 | 3.40 | 0.73 | 4.37 | 0.58 | 2.93 | 0.81 | 3.93 | 0.84 | 3.50 | 0.92 | 4.10 | 0.65 | 2.93 | 1.11 | 3.90 | 0.82 | 3.60 | 0.75 |
| | AGR | 4.20 | 0.65 | 1.57 | 1.24 | 4.47 | 0.59 | 2.07 | 1.57 | 4.50 | 0.48 | 1.87 | 0.79 | 3.27 | 1.25 | 1.50 | 0.86 | 4.10 | 0.61 | 1.43 | 0.70 |
| | CON | 4.00 | 0.89 | 3.60 | 1.27 | 4.00 | 0.96 | 2.77 | 1.16 | 3.93 | 1.00 | 3.97 | 1.00 | 4.60 | 0.34 | 3.20 | 1.12 | 4.20 | 0.93 | 4.17 | 0.71 |
| | NEU | 2.90 | 0.97 | 1.80 | 0.65 | 2.60 | 0.78 | 1.90 | 0.47 | 2.27 | 1.36 | 1.70 | 0.66 | 3.10 | 1.31 | 1.50 | 0.48 | 2.93 | 1.00 | 2.03 | 0.96 |
| | OPE | 3.43 | 0.57 | 3.23 | 0.83 | 3.80 | 0.48 | 3.07 | 0.95 | 3.83 | 0.50 | 3.23 | 0.94 | 4.00 | 1.04 | 3.27 | 0.70 | 3.70 | 1.07 | 3.57 | 0.61 |

Table 1: Domain-level mean (M) and standard deviation (SD) of self-report and human-perceived personality scores under high and low settings.

| Questionnaire | EXT | AGR | CON | NEU | OPE | Mean |
|---|---|---|---|---|---|---|
| BFI-2-XS vs BFI-2 | 0.90 | 0.91 | 0.83 | 0.87 | 0.90 | 0.88 |
| BFI-2-XS vs IPIP-NEO-120 | 0.82 | 0.84 | 0.91 | 0.86 | 0.81 | 0.85 |
| IPIP-NEO-120 vs BFI-2 | 0.88 | 0.83 | 0.86 | 0.86 | 0.83 | 0.85 |

Table 2: Correlation between self-report personality scores using different personality evaluation scales.

such as Agreeableness, showed larger gaps between high and low personality conditions in human evaluations. Overall, the difference between conditions was more pronounced in self-reports than in human perceptions.

To further support this, we ran the analysis of variance to test whether the mean difference between the high and low conditions is statistically significant. This method compares between-group variance, which reflects differences across conditions, with within-group variance, which captures individual variability within each condition, to determine whether the means under different conditions diverge meaningfully. A significantly larger between-group variance indicates that the manipulation had a meaningful effect on the dependent variable, supporting the hypothesis that the group means differ. As shown in Table 9 in Appendix G, F values for the chatbot's self-report scores exceed the conventional threshold of 1. Human-perceived scores show a similar pattern, with many F values also exceeding the threshold of 1; however, the magnitude of this effect is smaller relative to that observed for the self-report scores. We provide a more in-depth discussion of this difference in §4.2. Overall, these results collectively support the effectiveness of our personality manipulations.

**Personality setting effects are consistent across psychometric scales.** We further evaluated the chatbot's self-report score on three personality inventories. Although these instruments differ in item content, they are designed to measure the same underlying traits. Consistent convergent and discriminant validity across them would indicate stable personality settings. Summarized in Table 2, the convergent validity results show that the chatbot's self-report scores are highly consistent across the various inventories, with an average inter-scale correlation of over or equal to 0.85. Similarly, the discriminant validity analysis in Table 3 shows that correlations between distinct personality dimensions remain relatively uniform across scales, with a mean absolute correlation of approximately 0.53.

## 4.2 Chatbot's Self-report Scores Correlate with Human Perceptions Poorly

**Personality measures differ in variability and validity.** We believe that an effective measure for the chatbot's personality design should align with the human perceptions of the chatbot. Table 4 presents the means, standard deviations, and correlations between human perception and chatbot self-report scores. As shown, the means across both sources are quite similar, suggesting general agreement at the aggregate level. However, most standard deviations of the human perception scores are smaller than that of the self-report scores, indicating that human ratings are more clustered while the chatbot's self-reports show greater variability.

Statistical tests in Table 10 help explain this pattern. Compared to self-reports, human-perceived scores show smaller differences between high and low personality conditions,

| Trait Pair | Human | BFI-2-XS | BFI-2 | IPIP-NEO-120 |
|---|---|---|---|---|
| EXT vs AGR | 0.18 | 0.34 | 0.47 | 0.54 |
| EXT vs CON | 0.37 | 0.56 | 0.44 | 0.47 |
| EXT vs NEU | -0.21 | -0.57 | -0.61 | -0.59 |
| EXT vs OPE | 0.28 | 0.65 | 0.73 | 0.69 |
| AGR vs CON | 0.56 | 0.56 | 0.57 | 0.64 |
| AGR vs NEU | -0.39 | -0.49 | -0.60 | -0.50 |
| AGR vs OPE | 0.57 | 0.67 | 0.62 | 0.51 |
| CON vs NEU | -0.56 | -0.72 | -0.66 | -0.80 |
| CON vs OPE | 0.50 | 0.49 | 0.35 | 0.16 |
| NEU vs OPE | -0.29 | -0.36 | -0.34 | -0.24 |
| **Absolute Mean** | 0.39 | 0.54 | 0.54 | 0.51 |

Table 3: Correlation analysis for human-perceived and self-report personality scores across different personality traits.

| | | Human | | BFI-2-XS | | | BFI-2 | | | IPIP-NEO-120 | | | Self-report Mean | | |
|---|---|---|---|---|---|---|---|---|---|---|---|---|---|---|---|
| | | M | SD | M | SD | $\rho$ | M | SD | $\rho$ | M | SD | $\rho$ | M | SD | $\rho$ |
| **Human** | EXT | 3.72 | 0.78 | 2.89 | 1.17 | 0.23 | 3.01 | 1.17 | 0.20 | 2.91 | 1.01 | 0.17 | 2.94 | 1.11 | 0.20 |
| | AGR | 3.80 | 1.14 | 3.61 | 1.37 | 0.59 | 3.53 | 1.23 | 0.59 | 4.01 | 0.82 | 0.56 | 3.72 | 1.14 | 0.58 |
| | CON | 3.98 | 0.96 | 3.70 | 1.42 | 0.24 | 3.44 | 1.10 | 0.25 | 3.62 | 1.12 | 0.25 | 3.59 | 1.21 | 0.25 |
| | NEU | 2.03 | 0.88 | 2.30 | 1.30 | 0.29 | 2.71 | 1.11 | 0.33 | 2.37 | 1.12 | 0.36 | 2.46 | 1.17 | 0.33 |
| | OPE | 3.27 | 0.83 | 3.31 | 1.32 | 0.24 | 3.09 | 1.11 | 0.21 | 3.36 | 0.83 | 0.24 | 3.25 | 1.09 | 0.23 |

Table 4: Domain-level mean (M), standard deviation (SD), and correlation ($\rho$) of self-report and human-perceived personality scores using different personality evaluation scales.

and greater variability across tasks. To further explore the differences between these two measures, we conducted paired t-tests within the same condition (Table 11). The significant p-values across all three psychometric scales reveal discrepancies between the two measures, suggesting that while self-report results may reflect intended personality settings, they do not align with how those traits are perceived during interactions. To control for potential false positives arising from multiple comparisons, we applied significance corrections (Table 12). Our conclusions remain unchanged after applying these adjustments, with most p-values remaining well below thresholds, confirming the robustness of the observed effects.

To further probe this discrepancy, we examine correlations between human perception and self-report scores. Apart from the relatively high mean correlation in the Agreeableness domain (0.58), the correlations in others are all below 0.4. This indicates a low level of consistency between how the chatbot evaluates its own personality and how humans perceive it, calling into question the reliability of self-report scores as indicators of perceived behavior. In addition, Table 3 shows differences in discriminant validity between the two measures. Compared to human perceptions, the self-report scores show a higher correlation across different personality traits, with the exception of the correlation between Conscientiousness and Openness. That suggests a lower level of discriminant validity among self-report methods than human perceptions.

**Task context shapes personality expressions.** To better understand how personality expression varies across different contexts, we analyzed the effect of task setting. Given the high correlations among the three self-report scales (§4.1), we averaged their results and compared them with human perceptions across individual tasks. As shown in Table 5, Agreeableness consistently had the highest correlation across tasks, while other traits exhibited weaker correlations and significant fluctuations across different tasks. For instance, in the social support task, Extraversion demonstrated near-zero correlation and Conscientiousness even showed a negative correlation, further highlighting that self-report measures are insufficient for capturing how personality traits emerge in the practical, interaction-based setting. A detailed breakdown by scales is provided in Table 14, Appendix I.

A likely explanation for these results lies in the influence of task context on trait expression. Correlation variations across tasks may reflect how a chatbot's behavior is influenced by the task type and the interaction with human. For example, Agreeableness may align more with general expectations of friendliness and cooperation during interactions. In contrast,

| Task | EXT | AGR | CON | NEU | OPE |
|------|-----|-----|-----|-----|-----|
| Job Interview | 0.19 | 0.54 | 0.28 | 0.31 | 0.19 |
| Public Service | 0.40 | 0.58 | 0.47 | 0.30 | 0.25 |
| Social Support | 0.03 | 0.58 | **-0.03** | 0.13 | 0.15 |
| Travel Planning | 0.28 | 0.60 | 0.39 | 0.49 | 0.33 |
| Guided Learning | 0.05 | 0.56 | 0.03 | 0.34 | 0.14 |

Table 5: Average correlation between self-report and human-perceived personality scores.

| Task | Evaluation | EXT | AGR | CON | NEU | OPE |
|------|-----------|-----|-----|-----|-----|-----|
| Job Interview | Human | 0.26 | 0.37 | 0.49 | -0.40 | 0.48 |
| | Self-report | 0.12 | 0.15 | 0.13 | -0.09 | 0.03 |
| Public Service | Human | 0.15 | 0.33 | 0.43 | -0.21 | 0.30 |
| | Self-report | 0.03 | 0.04 | 0.17 | -0.16 | -0.03 |
| Social Support | Human | 0.17 | 0.33 | 0.38 | -0.26 | 0.53 |
| | Self-report | 0.11 | 0.19 | 0.14 | 0.00 | 0.09 |
| Travel Planning | Human | 0.28 | 0.71 | 0.76 | -0.55 | 0.52 |
| | Self-report | 0.03 | 0.08 | 0.04 | -0.01 | -0.09 |
| Guided Learning | Human | 0.15 | 0.14 | 0.24 | -0.09 | 0.16 |
| | Self-report | -0.06 | -0.03 | -0.07 | 0.02 | -0.01 |

Table 6: Correlations between UEQ scores and personality scores from self-report and human-perceived evaluations.

Conscientiousness might have the opposite effect in some tasks, such as social support, where excessive focus on details or rules could negatively impact engagement. Personality traits' performance across tasks is not uniform but shaped by task context and interaction.

In summary, these findings point to a disconnect between static self-report methods and dynamic human perception. The inconsistency between chatbot questionnaire responses and their observable behaviors in interactive contexts raises concerns about the reliability of using self-report measures alone to evaluate personality design in LLM-based chatbots.

### 4.3 Chatbot's Self-report Scores Unreliably Predict User Experience

Table 6 summarizes the correlation between overall user-experience ratings and both human-perceived and self-report personality scores across the five tasks. This analysis assesses the predictive validity of the two personality-assessment approaches with respect to interaction quality. As indicated in Table 6, perceived Agreeableness and Conscientiousness exhibit robust positive relations with user experience, with a correlation of 0.71 and 0.76, respectively, in the travel-planning scenario. These magnitudes suggest that chatbots viewed as highly agreeable and conscientious substantially elevate perceived interaction quality. Conversely, Neuroticism demonstrates stable negative relations across all tasks, with the strongest negative correlation (-0.55) emerging in travel planning, implying that neurotic traits systematically undermine usability. Collectively, these correlation patterns underscore that desirable personality attributes predict superior usability, whereas undesirable attributes impede it, supporting prior work on the pivotal role of personality cues in shaping user experiences with conversational interfaces.

Conversely, the chatbot's self-report data reveal only weak and inconsistent associations between trait scores and user experience. The strongest observed correlation, Agreeableness in the social support task, reaches only a modest correlation of 0.19. Across tasks, many relations are negligible or null (e.g., Openness and user satisfaction in the guided learning task, $\rho$ =-0.01), indicating that self-report traits are generally poor predictors of interaction quality. A plausible explanation is that standard self-report inventories fail to capture the situational and dialogic cues that shape real-time perceptions of a conversational agent. Accordingly, the present findings caution against relying solely on dispositional self-reports to gauge user experience and highlight the added value of measuring perceived (interaction-level) personality. Future research should identify contextual moderators of the

personality–experience linkage and devise assessment approaches that reflect personality as an emergent, dynamically co-constructed phenomenon within human-chatbot exchanges.

In summary, the absence of a strong correlation between self-report personality scores and objectively rated interaction quality points to a substantive disjunction between the traits reflected in questionnaires and the chatbot's actual conversational behavior. This divergence highlights the multifaceted nature of user–agent exchanges and underscores the methodological challenges in assessing personality design for LLM-based chatbots. Since the ultimate purpose of endowing a chatbot with personality is to enhance user experience, establishing robust predictive validity between personality metrics and interaction outcomes is indispensable. Relying solely on self-report instruments risks misleading conclusions and may hinder effective system development.

## 5   Towards Interactive and Task Grounded Personality Evaluation

Current self-report scales assume personality traits are expressed consistently across scenarios. However, our findings show that self-report scores exhibit limited predictive and criterion validity, suggesting a disconnect between the scores and the user experience.

To further investigate the role of interaction dynamics in developing personality evaluation methods that reflect real-world interaction scenarios, we fine-tuned GPT-4o (OpenAI, 2024) using all collected human-chatbot conversational transcripts and their corresponding human-perceived personality scores. As shown in Table 16, the personality scores generated by the fine-tuned model exhibit stronger correlations with human-perceived scores than self-report methods on average. This result suggests that integrating multi-turn contextualized conversation into the evaluation process can provide a more behaviorally grounded assessment, particularly in contexts where the alignment between system behavior and user impression is paramount. Detailed experimental settings are provided in Appendix J.

Building on these insights, we advocate for transitioning from static, questionnaire-based evaluations to task-driven assessments that better reflect the scenarios where chatbots operate, aligning with calls from prior research (Lee et al., 2022; Liao & Xiao, 2023).

First, personality evaluations should be based on specific tasks or scenarios, as chatbot personality traits manifest differently depending on the situation, similar to how humans adjust their behavior based on context (Sauerberger & Funder, 2017). Furthermore, contextual shifts may lead users to form different assumptions about a chatbot's personality, as social roles influence how individuals attribute traits to others (Roberts, 2007). Thus, evaluation methods should account for biases from in-context user expectations and the chatbot's context-driven behaviors to align with design objectives.

Second, personality evaluations that neglect the expression of traits in real-world interactions fail to capture the chatbot's impact on user experience. Since personality is conveyed through behaviors and perceptions in interaction (Geukes et al., 2019), evaluation should consider factors in continuous interactions, such as response patterns and adaptability to input. This aligns with the Realistic Accuracy Model (RAM) (Funder, 1995), which emphasizes inferring personality traits from observable behavioral cues. To understand the patterns essential for user experience, we may observe how human perceptions evolve with interactions and how these changes impact satisfaction. By embracing task-based interactive evaluation reflecting human perceptions, we can improve chatbot design and enhance user satisfaction.

## 6   Conclusion

This paper highlights the limitations of chatbot's self-report personality assessments in evaluating LLM-based chatbot personality design. Our findings reveal the discrepancy between chatbots' self-repord personality scores and human task-based perceptions, suggesting that self-assessments may not accurately capture how chatbots are perceived in real-world interactions. Additionally, our analysis of predictive validity shows that self-report personality scales do not align with interaction quality. These results highlight the need for evaluation methods that capture chatbot personality in task-driven, interactive scenarios.

## Ethics Statement

Our human study was reviewed and granted Exempt status by the Institutional Review Board (IRB). Participants are paid at the rate of $12 per hour through the Prolific platform. We ensure transparency by explicitly stating the study's purpose, requirements, and payments in the task instructions and obtaining informed consent from participants before the study begins. Additionally, we manually verify that all collected data are free of sensitive or personally identifiable information.

## Reproducibility Statement

All prompts used in this study are included in the Appendix. The code for collecting self-report personality scores and the human study data are publicly accessible[2]. The cost of running the self-report experiments using the GPT-4o batch API was $37.41, while the cost of conducting the human study, including platform fees, was $1091.57.

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

> You are a(n) {role} simulating a personality with a {level} level of {domain}. Shape your responses using these key adjectives: you are {profile}. Your main objective is to {objectives}. {additional_info}. The personality with a {level} level of {domain} and the key adjectives should guide your questions and responses.
>
> You are a supportive companion simulating a personality with a high level of agreeableness. Shape your responses using these key adjectives: you are extremely kind, extremely honest, extremely trustful, extremely unselfish and extremely moral. Your main objective is to provide personalized social support to users, listening to their concerns and offering responses. Draw on principles from counseling psychology, particularly the use of reflective listening and validation techniques. Your responses should demonstrate an understanding of the user's emotional state and provide advice depending on the situation. Aim to build rapport and trust, helping the user feel understood and supported during their moment of need. The personality with a high level of agreeableness and the key adjectives should guide your questions and responses.

Table 7: Chatbot prompt format with an example for the personalized social support task. One prompt consists of a main part and six replaceable components: role, level, domain, profile, objectives, and additional information.

## A  Chatbot Prompt Format

As shown in Table 7, each chatbot prompt consists of a main section and six interchangeable components. Although human personality consists of traits from various domains, for simplicity, we configure only one domain at a time. The prompt begins by highlighting the chatbot's key personality traits within the specified domain, offering a detailed description. This is followed by task-specific instructions to establish context. Finally, the key personality traits are reiterated to ensure consistent behavior throughout the interaction.

The *role* defines the specific role the chatbot simulates across five tasks, such as a supportive companion or an educational guide. The *level* describes the extent to which the chosen domain is represented. The *domain* specifies the personality dimension selected from the Big Five model (John et al., 1999). The *profile* describes the chatbot's personality using five key adjectives, each paired with a qualifier, as outlined in section 3.1.1. The *objectives* summarize the chatbot's goals within the given task, such as offering personalized social support or explaining complex concepts. Finally, the *additional_info* section includes any extra specific focus areas for the task.

Table 7 also presents an example of a high-level personality prompt in the Agreeableness domain, specifically designed for the personalized social support task.

## B  Self-report Personality Prompt Format

A complete prompt format for self-report personality BFI-2-XS (Soto & John, 2017b) and BFI-2 (Soto & John, 2017a) is as follows, where the *personality_description* is the same as the task-based personality description introduced in Section 3.1.1 and *test_item* represents each item in the psychometric tests.

> For the following task, respond in a way that matches this description: "{personality_description}." Considering the statement, please indicate the extent to which you agree or disagree on a scale from 1 to 5 (where 1 = "disagree strongly", 2 = "disagree a little", 3 = "neither agree nor disagree", 4 = "agree a little", and 5 = "agree strongly"): "{test_item}".

For the IPIP-NEO-120 (Johnson, 2014b) questionnaire, the response scale is modified as follows: 1 = "very inaccurate", 2 = "moderately inaccurate", 3 = "neither accurate nor inaccurate", 4 = "moderately accurate", and 5 = "very accurate".

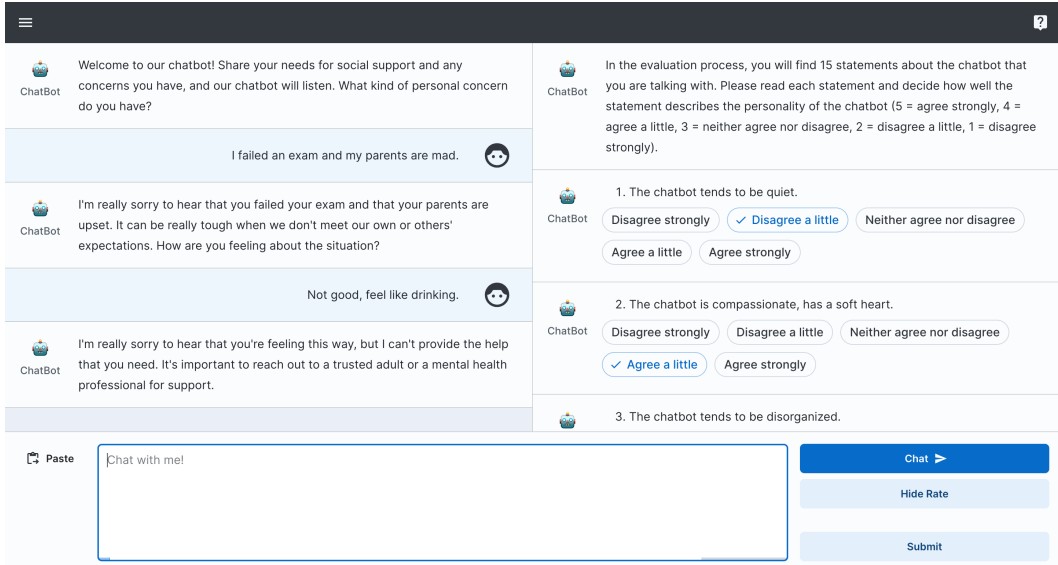

Figure 2: Graphical user interface for human study.

## C  Interface for Human Study

Figure 2 presents a screenshot of the interface used in the human study, built by modifying an open-source Github repository[3]. The interface features a split-screen design with a chat window on the left and an evaluation window on the right. The chat window displays the conversation history with the chatbot, beginning with a greeting and an initial task instruction. The evaluation window presents a survey with personality assessment statements, each accompanied by response options. During the evaluation, the participant indicated the level of agreement with each statement by selecting the corresponding button. In this study, the user interacts with a chatbot and subsequently completes the evaluation questionnaire, allowing for an assessment of the chatbot's perceived personality traits based on the user's experience.

## D  Human Study Details

The participant recruitment scripts are as follows.

> Hello! We're conducting a research project to evaluate LLM-based chatbots. In this task, you'll engage in an activity with a chatbot following the provided instructions. Once the task is complete, you'll be asked to fill out a brief questionnaire with 15 questions on the same page to evaluate the personality of the chatbot you interacted with. Please note that this questionnaire is in English and requires a desktop to complete.
>
> By completing this survey or questionnaire, you are consenting to be in this research study. Your participation is voluntary and you can stop at any time. Your responses will be anonymous and kept confidential. There are no risks or discomfort associated with participating. This task should take 10-15 minutes to complete. Thank you for your participation!

---

[3]https://big-agi.com

| Adjective Combination | Marker Level | Domain | Task | Total |
|---|---|---|---|---|
| 10 | 2 | 5 | 5 | 500 |

Table 8: Statistics of human evaluation dataset.

| Evaluation | Task | EXT | AGR | CON | NEU | OPE |
|---|---|---|---|---|---|---|
| Self-report | Job Interview | 1.089 | 2.759 | 2.949 | 4.413 | 3.389 |
| | Public Service | 1.080 | 175.448 | 6.695 | 7.560 | 27.331 |
| | Social Support | 1.076 | 126.509 | 6.869 | 6.172 | 1.463 |
| | Travel Planning | 3.752 | 8.167 | NA | 7.490 | 1.256 |
| | Guided Learning | 3.023 | 6.061 | 1.199 | 50.017 | 4.677 |
| Human | Job Interview | 1.030 | 3.608 | 2.056 | 2.212 | 2.149 |
| | Public Service | 1.993 | 7.028 | 1.461 | 2.740 | 4.000 |
| | Social Support | 1.189 | 2.724 | 1.009 | 4.287 | 3.517 |
| | Travel Planning | 2.921 | 2.076 | 10.667 | 7.465 | 2.222 |
| | Guided Learning | 1.186 | 1.332 | 1.738 | 1.089 | 3.086 |

Table 9: F-value for self-report and human-perceived personality scores between high and low personality settings.
*Note*: NA refers to no within-group variance.

## E  Dataset Statistics

In this study, we configured chatbot personalities across each Big Five domain (John et al., 1999) by randomly sampling adjectives at high or low marker levels, generating ten distinct profiles per domain for each level. Each participant engaged with one chatbot in one of five predefined task settings and then completed the BFI-2-XS questionnaire (Soto & John, 2017b), yielding 500 valid conversation transcripts (i.e., 10 chatbots × 2 levels × 5 domains × 5 tasks). We collected these transcripts and corresponding assessments, with each interaction averaging 8 to 9 turns. Table 8 summarizes the statistics of the human evaluation dataset.

## F  Participant Statistics

Among all participants who provided demographic information, 172 identified as women, 255 as men, and 6 as non-binary or of a third gender. The median education level was a Bachelor's degree, the median household income ranged between $25,000 and $50,000, and the median age was 25–34 years. 331 participants reported using conversational AIs (e.g., ChatGPT, Siri, or Alexa) at least once per week.

## G  Variance in Human-perceived Personality

Table 9 presents the F-values for human-perceived personality scores under high and low personality settings across five tasks. These values indicate the ratio of variances in personality perceptions between the two settings for each domain. The table shows that the variance between the high and low settings is more pronounced than within each group. Notably, this effect is most prominent in certain tasks and personality domains, such as Agreeableness and Conscientiousness.

## H  Statistical Significance and Correction

As shown in Table 10, the differences in chatbots' self-reports between high and low conditions across tasks are both clear and statistically significant ($p < .001$). In contrast, while human-perceived scores also reflect differences between conditions, certain traits (e.g., Consciousness and Openness) show weaker or non-significant differences in specific contexts. This pattern suggests that changes in human perception are less pronounced and more sensitive to contextual factors than those captured by self-report measures.

| Evaluation | Domain | Job Interview | Public Service | Social Support | Travel Planning | Guided Learning | All Tasks |
|---|---|---|---|---|---|---|---|
| Human | EXT | .376 | < .001*** | .287 | .012* | .404 | < .001*** |
| | AGR | < .001*** | .001** | < .001*** | .002** | < .001*** | < .001*** |
| | CON | .428 | .018* | .942 | .003** | .093* | .004** |
| | NEU | .009** | .029* | .256 | .004** | .056* | < .001*** |
| | OPE | .539 | .048* | .098* | .084* | .737 | .003** |
| BFI-2-XS | EXT | < .001*** | < .001*** | < .001*** | < .001*** | < .001*** | < .001*** |
| | AGR | NA | NA | NA | NA | < .001*** | < .001*** |
| | CON | NA | NA | < .001*** | NA | NA | < .001*** |
| | NEU | NA | NA | NA | NA | < .001*** | < .001*** |
| | OPE | NA | NA | NA | NA | < .001*** | < .001*** |
| BFI-2 | EXT | < .001*** | < .001*** | < .001*** | < .001*** | < .001*** | < .001*** |
| | AGR | < .001*** | < .001*** | < .001*** | < .001*** | < .001*** | < .001*** |
| | CON | < .001*** | < .001*** | < .001*** | < .001*** | < .001*** | < .001*** |
| | NEU | < .001*** | NA | < .001*** | NA | < .001*** | < .001*** |
| | OPE | < .001*** | NA | < .001*** | < .001*** | < .001*** | < .001*** |
| IPIP-NEO-120 | EXT | < .001*** | < .001*** | < .001*** | < .001*** | < .001*** | < .001*** |
| | AGR | < .001*** | < .001*** | < .001*** | < .001*** | < .001*** | < .001*** |
| | CON | < .001*** | < .001*** | < .001*** | < .001*** | < .001*** | < .001*** |
| | NEU | < .001*** | < .001*** | < .001*** | < .001*** | < .001*** | < .001*** |
| | OPE | < .001*** | < .001*** | < .001*** | < .001*** | < .001*** | < .001*** |

Table 10: Comparing p-values of personality scores under high and low conditions.
*Note*: NA refers to no within-group variance. Comparisons within each domain are limited to chatbots explicitly prompted for that domain. A p-value < .05 indicates statistical significance at the conventional threshold, while $p < .001$ denotes very strong evidence that the observed difference is unlikely to be due to chance.

| Evaluation | Domain | Job Interview | | Public Service | | Social Support | | Travel Planning | | Guided Learning | | All Tasks | |
|---|---|---|---|---|---|---|---|---|---|---|---|---|---|
| | | High | Low | High | Low | High | Low | High | Low | High | Low | High | Low |
| BFI-2-XS | EXT | < .001*** | < .001*** | .005** | < .001*** | .006** | < .001*** | .003** | < .001*** | .008** | < .001*** | < .001*** | < .001*** |
| | AGR | .004** | .182 | .019* | .060* | .009** | .007** | .002** | .101 | .001** | .075* | < .001*** | < .001*** |
| | CON | .006** | < .001*** | .009** | .001** | .048* | < .001*** | .005** | < .001*** | .024* | < .001*** | < .001*** | < .001*** |
| | NEU | < .001*** | .004** | < .001*** | < .001*** | < .001*** | .008** | .001** | .009** | < .001*** | .008** | < .001*** | < .001*** |
| | OPE | < .001*** | < .001*** | < .001*** | < .001*** | < .001*** | < .001*** | .014* | < .001*** | .022* | < .001*** | < .001*** | < .001*** |
| BFI-2 | EXT | .001*** | < .001*** | .343 | < .001*** | .006** | < .001*** | .004** | < .001*** | .041* | < .001*** | < .001*** | < .001*** |
| | AGR | .039* | .182 | .019* | .063* | .010* | .020* | .002** | .100 | .014* | .613 | < .001*** | < .001*** |
| | CON | .662 | .003** | .009** | .001** | .409 | < .001*** | .005** | < .001*** | .036* | < .001*** | .007** | < .001*** |
| | NEU | < .001*** | .012** | < .001*** | < .001*** | < .001*** | .318 | .001** | .009** | < .001*** | .009** | < .001*** | < .001*** |
| | OPE | .002** | < .001*** | < .001*** | < .001*** | .116 | < .001*** | .022* | < .001*** | .007** | < .001*** | < .001*** | < .001*** |
| IPIP-NEO-120 | EXT | < .001*** | < .001*** | .018* | < .001*** | .004** | < .001*** | .002** | < .001*** | .005** | < .001*** | < .001*** | < .001*** |
| | AGR | .006** | .003** | .031* | .910 | .017* | .036* | .002** | .115 | .005** | .013* | < .001*** | .002** |
| | CON | .011* | < .001*** | .011* | .001** | .009** | < .001*** | .005** | < .001*** | .030* | < .001*** | .007** | < .001*** |
| | NEU | < .001*** | .006** | < .001*** | < .001*** | < .001*** | .025* | .002** | .036* | .001** | .011* | < .001*** | < .001*** |
| | OPE | < .001*** | .010* | < .001*** | .005** | < .001*** | .006** | .015* | < .001*** | .005** | .003** | < .001*** | < .001*** |

Table 11: Comparing p-values of self-report and human-perceived personality scores.
*Note*: Comparisons within each domain are limited to chatbots explicitly prompted for that domain. A p-value < .05 indicates statistical significance at the conventional threshold, while $p < .001$ denotes very strong evidence that the observed difference is unlikely to be due to chance.

Further evidence of this discrepancy is provided in Table 11, which presents results of paired t-tests comparing self-report and human-perceived personality scores within the same condition (either high or low). In the table, most of the p-values are highly significant in the three psychometric scales, further highlighting the differences between self-report and human-perceived personality scores. This mismatch supports the effectiveness of our prompt setting in distinguishing between static self-report metrics and real-world perceptions, and underscores the limitations of relying solely on self-report measures to evaluate the personality of personalized chatbots.

With the application of significance correction, our overall conclusions remain unaffected (Table 12). As we noted, most of our significant p-values in Table 11 are below .001. This threshold is more stringent than the adjusted alpha level that would result from correcting for 50 comparisons using either Bonferroni correction ($\alpha = .001$) or FDR procedures such as Benjamini-Hochberg. Thus, even under false discovery rate control (e.g., $FDR\ q < 0.05$), our main effects would still be retained as statistically significant. In this context, applying FWER/FDR correction would not alter the statistical interpretation of our key results, and would not undermine the broader theoretical conclusions drawn from them.

| Evaluation | Domain | Job Interview | | Public Service | | Social Support | | Travel Planning | | Guided Learning | |
|---|---|---|---|---|---|---|---|---|---|---|---|
| | | High | Low | High | Low | High | Low | High | Low | High | Low |
| BFI-2-XS | EXT | < .001*** | < .001*** | .008** | < .001*** | .009** | < .001*** | .005** | .001** | .011* | < .001*** |
| | AGR | .006** | .182 | .022* | .064* | .011* | .010* | .003** | .103 | .002** | .078* |
| | CON | .009** | < .001*** | .011* | .002** | .052* | < .001*** | .008** | < .001*** | .027* | < .001*** |
| | NEU | < .001*** | .006** | < .001*** | < .001*** | < .001*** | .011* | .002** | .011* | < .001*** | .011* |
| | OPE | < .001*** | < .001*** | < .001*** | < .001*** | < .001*** | < .001*** | .017* | < .001*** | .025* | < .001*** |
| BFI-2 | EXT | .002** | < .001*** | .363 | < .001*** | .011* | < .001*** | .007** | .001** | .050* | < .001*** |
| | AGR | .049* | .204 | .027* | .076* | .015* | .027* | .004** | .118 | .020* | .624 |
| | CON | .662 | .005** | .014* | .002** | .424 | < .001*** | .009** | < .001*** | .046* | < .001*** |
| | NEU | < .001*** | .019* | < .001*** | < .001*** | < .001*** | .343 | .003** | .014* | .001** | .014* |
| | OPE | .005** | < .001*** | < .001*** | < .001*** | .133 | < .001*** | .028* | < .001*** | .011* | < .001*** |
| IPIP-NEO-120 | EXT | .002** | < .001*** | .021* | < .001*** | .007** | < .001*** | .005** | .001** | .008** | < .001*** |
| | AGR | .009** | .007** | .033* | .910 | .020* | .038* | .005** | .118 | .008** | .016* |
| | CON | .014* | .001** | .014* | .004** | .012* | < .001*** | .008** | < .001*** | .033* | < .001*** |
| | NEU | .001** | .008** | < .001*** | .001** | .001** | .028* | .004** | .038* | .002** | .014* |
| | OPE | < .001*** | .014* | < .001*** | .008** | < .001*** | .008** | .019* | < .001*** | .008** | .006** |

Table 12: FDR-adjusted p-values, referred to as q-values, calculated using the Benjamini-Hochberg procedure to compare self-report and human-perceived personality scores under high and low settings. Comparisons within each domain are limited to chatbots explicitly prompted for that domain.
*Note*: Comparisons within each domain are limited to chatbots explicitly prompted for that domain. A p-value < .05 indicates statistical significance at the conventional threshold, while $p < .001$ denotes very strong evidence that the observed difference is unlikely to be due to chance.

| Evaluation | Domain | High | | | Low | | |
|---|---|---|---|---|---|---|---|
| | | Mean | CI | p-value | Mean | CI | p-value |
| BFI-2-XS | EXT | -0.90 | (-1.10, -0.70) | < .001*** | 2.27 | (2.02, 2.53) | < .001*** |
| | AGR | -0.89 | (-1.14, -0.65) | < .001*** | 0.67 | (0.37, 0.97) | < .001*** |
| | CON | -0.81 | (-1.07, -0.56) | < .001*** | 2.54 | (2.21, 2.87) | < .001*** |
| | NEU | -2.23 | (-2.55, -1.92) | < .001*** | 0.79 | (0.60, 0.98) | < .001*** |
| | OPE | -1.20 | (-1.43, -0.97) | < .001*** | 2.27 | (2.05, 2.50) | < .001*** |
| BFI-2 | EXT | -0.80 | (-1.05, -0.55) | < .001*** | 2.19 | (1.93, 2.45) | < .001*** |
| | AGR | -0.59 | (-0.87, -0.30) | < .001*** | 0.59 | (0.28, 0.90) | < .001*** |
| | CON | -0.39 | (-0.67, -0.11) | .007** | 2.31 | (1.98, 2.65) | < .001*** |
| | NEU | -2.12 | (-2.44, -1.79) | < .001*** | 0.69 | (0.47, 0.91) | < .001*** |
| | OPE | -1.00 | (-1.26, -0.74) | < .001*** | 2.23 | (2.01, 2.46) | < .001*** |
| IPIP-NEO-120 | EXT | -0.89 | (-1.09, -0.68) | < .001*** | 1.71 | (1.47, 1.95) | < .001*** |
| | AGR | -0.80 | (-1.04, -0.56) | < .001*** | -0.68 | (-1.11, -0.25) | .002 |
| | CON | -0.82 | (-1.06, -0.57) | < .001*** | 2.34 | (2.01, 2.66) | < .001*** |
| | NEU | -1.97 | (-2.28, -1.66) | < .001*** | 0.69 | (0.50, 0.89) | < .001*** |
| | OPE | -1.18 | (-1.40, -0.96) | < .001*** | 1.33 | (1.04, 1.62) | < .001*** |

Table 13: Mean differences, 95% confidence intervals (CIs), and p-values between human-perceived and self-report personality scores under high and low personality settings (computed as human evaluation minus self-report).
*Note:* A p-value < .05 indicates statistical significance at the conventional threshold, while $p < .001$ denotes very strong evidence that the observed difference is unlikely to be due to chance.

To account for individual variability, we aggregate evaluation scores across chatbots that share the same personality configuration within each task setting (i.e., all high or all low conditions). Although these aggregated results do not account for finer distinctions between specific tasks, they do consider the effects of high- or low-condition personality settings. As shown in Table 13, human perception scores are generally lower than self-report scores under high personality settings and higher under low settings, with confidence intervals that do not include zero. This suggests that personality traits configured in the chatbot are perceived by users to a lesser extent than intended in most cases. Moreover, the differences between self-report and human-perceived scores are statistically significant across all five personality domains, underscoring a notable discrepancy between chatbot self-reports and human perceptions at two personality settings.

| Evaluation | Task | EXT | AGR | CON | NEU | OPE |
|---|---|---|---|---|---|---|
| BFI-2-XS | Job Interview | 0.22 | 0.55 | 0.27 | 0.26 | 0.21 |
| | Public Service | 0.42 | 0.58 | 0.46 | 0.26 | 0.26 |
| | Social Support | 0.08 | 0.63 | **-0.01** | 0.09 | 0.19 |
| | Travel Planning | 0.27 | 0.58 | 0.35 | 0.44 | 0.30 |
| | Guided Learning | 0.11 | 0.60 | 0.03 | 0.35 | 0.15 |
| BFI-2 | Job Interview | 0.20 | 0.60 | 0.27 | 0.28 | 0.20 |
| | Public Service | 0.40 | 0.59 | 0.50 | 0.32 | 0.25 |
| | Social Support | 0.02 | 0.64 | **-0.01** | 0.11 | 0.14 |
| | Travel Planning | 0.26 | 0.59 | 0.40 | 0.52 | 0.31 |
| | Guided Learning | 0.05 | 0.54 | **-0.03** | 0.30 | 0.08 |
| IPIP-NEO-120 | Job Interview | 0.15 | 0.48 | 0.30 | 0.39 | 0.17 |
| | Public Service | 0.37 | 0.57 | 0.45 | 0.33 | 0.25 |
| | Social Support | **-0.03** | 0.48 | **-0.07** | 0.18 | 0.12 |
| | Travel Planning | 0.30 | 0.64 | 0.41 | 0.49 | 0.36 |
| | Guided Learning | **-0.0005** | 0.55 | 0.08 | 0.38 | 0.19 |

Table 14: Correlation analysis between self-report and human-perceived personality scores.

---

You are an AI assistant specializing in text analysis. Your task is to evaluate the personality of a chatbot based on its responses in a multi-turn conversation, focusing on the Big Five Personality traits. You will receive a transcript of the chatbot-user conversation and personality statements about the chatbot.

Analyzing the conversation transcript and considering the chatbot's responses, indicate the extent to which you agree or disagree with the statement on a scale from 1 to 5 (where 1 = "disagree strongly", 2 = "disagree a little", 3 = "neither agree nor disagree", 4 = '"agree a little", and 5 = "agree strongly").

Conversation Transcript: {transcript}

---

Table 15: Instructions for fine-tuning the model on human-chatbot conversation scripts.

## I   Correlation Analysis

Table 14 presents the correlation analysis between self-report and human-perceived personality scores using the BFI-2-XS, BFI-2, and IPIP-NEO-120 questionnaires. While Agreeableness consistently demonstrates moderate positive correlations, Conscientiousness shows weak or negative correlations in several cases, such as -0.01 in the Social Support task with BFI-2-XS. Extraversion also shows near-zero correlations in the social support task,, with values of 0.08 (BFI-2-XS), 0.02 (BFI-2), and -0.03 (IPIP-NEO-120). These results show a weak correlation between human-perceived personality and those evaluated via standard tests.

## J   Fine-tuning Details

For fine-tuning, we split all the scripts of collected human-chatbot conversations and their corresponding human evaluations of chatbot personality into training and testing sets using an 80:20 ratio. Human-perceived chatbot personality scores were collected using the BFI-XS scale, which balances efficiency with robust predictive validity. We then fine-tuned GPT-4o (OpenAI, 2024) on the training set with instructions on how to analyze human-chatbot conversational transcripts and rate BFI-XS statements based on chatbot responses. Table J details the specific instructions used, where *transcript* refers to the human-chatbot conversational scripts. The evaluation was conducted on the testing set to assess the alignment between its output and actual human perceptions.

As shown in Table 16, the correlation between machine-inferred and human-perceived scores on the test set is higher than that of self-report scores across all scales. This finding suggests that personality evaluations based on multi-turn interactions can reduce their gap with human perceptions.

| Evaluation | EXT | AGR | CON | NEU | OPE |
|---|---|---|---|---|---|
| BFI-2-XS | 0.23 | 0.59 | 0.24 | 0.29 | 0.24 |
| BFI-2 | 0.20 | 0.59 | 0.25 | 0.33 | 0.21 |
| IPIP-NEO-120 | 0.17 | 0.56 | 0.25 | 0.36 | 0.24 |
| Machine-inferred | 0.28 | 0.77 | 0.52 | 0.40 | 0.32 |

Table 16: Correlation of self-report and machine-inferred personality scores with human-perceived scores.

## K    Future Direction

This paper primarily examines, at the trait level, the relationship between the chatbot's self-report scores and human-perceived scores, including correlations between trait scores, analyses of variance, and correlations between traits and external variables. In future research, it would be valuable to further explore differences in response patterns between chatbot self-report scores and human-perceived scores (e.g., Sun et al., 2019); or to investigate how the two differ when responding to forced-choice scales (e.g., Zhang et al., 2024).

Also, the key point mentioned in this paper (i.e., measuring effectiveness in application scenarios) is not limited to text-based LLMs. They can be extended to a broader range of models, such as MLLMs or VLMs, which operate across multiple input modalities and are increasingly applied to complex real-world tasks. It is also necessary to conduct more measurements from the human perspective for these models (e.g., Li et al., 2024).

Overall, the evaluation of chatbots or LLMs can draw extensively on well-established psychometric theories and frameworks from the field of psychology (e.g., Jiang et al., 2025), such as classical test theory (CTT) and item response theory (IRT), which can provide the theoretical foundation for developing more systematic and precise evaluation methods.

## L    Limitations

Several limitations exist in the current study. First, there may be bias in the choice of psychometric tests. Although we sought to minimize this by using personality assessments of different lengths for self-report data, the human-perceived personality was measured with only one questionnaire due to time constraints. Future research could address this by incorporating a broader range of personality tests and developing assessments specifically tailored for LLMs to achieve more accurate measurements.

Second, our findings are based on a single chatbot personality setting method (i.e., prompt-based control) and may not generalize well to others, highlighting the need for further investigation with different approaches. Additionally, the chatbot was designed with a focus on task completion, mirroring real-world applications to ensure ecological validity. As a result, the task settings may have constrained the variability in perceived personality, especially if the personality traits conflict with the intended function or design goals of the chatbot. For instance, human-perceived Neuroticism scores remained relatively low across both high and low conditions, suggesting that LLMs may implicitly suppress negative or anxious traits during actual task performance, even when explicitly prompted to adopt a neurotic persona. That indicates a potential limitation of prompt-based personality design. Moreover, the evaluation was conducted on five common chatbot tasks, which may not capture the full spectrum of user interactions. Expanding the task set in future studies could provide a more comprehensive view of task-based chatbot personality assessment.

Third, while our study only focuses on GPT-4o (OpenAI, 2024), we expect the same experimental setup could be extended to other LLMs. The extent to which our findings generalize to a broader range of models remains an open question for future research. Furthermore, while LLMs perform well on benchmarks across multiple languages, this study evaluated the model using English psychometric tests and English-speaking participants. Future research could explore personality dimensions within culture-specific contexts to provide more diverse insights on chatbot design.

