# OpenReview forum: "Can LLM "Self-report"?: Evaluating the Validity of Self-report Scales in Measuring Personality Design in LLM-based Chatbots"
_colmweb.org/COLM/2025/Conference — COLM 2025_

### Official Review · Reviewer_2DCa · 2025-05-09

**Rating:** 10
**Confidence:** 5
**Ethics Flag:** 1

**Summary:**

This excellent and well written paper provides a methodology and design of evaluating  the chatbot self reported personality and human perceived ones. They demonstrate through their work with GPT-4o that this approach to self reporting a chatbot’s personality is almost always similar to human perception of its personality. The authors use very well established and stable personality tests to approach the task.

**Reasons To Accept:**

1. Very pertinent problem statement with conclusions mirrors reviewers own observations on other independent datasets
2. Very good coverage of old and new relevant references and pertinent study
3. Very good representative figures
4. They provide a very good detailed chatbot design methodology
5. Setting the temperature to 0 in GPT-4o allowed repeatable experiments
6. Personality Design Evaluation is very sound with use of stable well established psychometric tests
7. The Construct Validity Evaluation is sound and the results very illuminating
8. The authors in the appendix provide very pertinent future work as well

**Reasons To Reject:**

None that I can come across

---

> ### Author Response · Authors · 2025-06-02
> **Summary**
>
> We sincerely thank reviewer 2DCa for the encouraging feedback. We appreciate your recognition of our problem statement and study design. We would also like to take this opportunity to emphasize that our work contributes to evaluating the validity of using self-report questionnaires in LLM evaluation. We offer a systematic method to evaluate the validity of such methods and our results demonstrate the limitations of using such scales in measuring LLM’s personality design in realistic interactions. This mismatch highlights the limitations of relying solely on self-report methods for LLM-based chatbot evaluation and underscores the need for interactive, human-centered evaluation approaches. Thank you again for your encouraging review.

---

### Official Review · Reviewer_ezgD · 2025-05-12

**Rating:** 7
**Confidence:** 4
**Ethics Flag:** 1

**Summary:**

This paper addresses the problem of evaluating chatbot personality, specifically questioning the validity of self-reported personality in the context of large language model based systems. Two sets of validity: intrinsic and extrinsic.

The experimental results revealed a discrepancy between self-reported personality and perception of human evaluations of chat personality. The results also showed a lack of alignment between self-reported personality and interaction quality.

**Questions To Authors:**

What is the statistical significance of the reported results?

>> The authors have addressed this question already

**Reasons To Accept:**

The topic is important and relevant to the research community. This work can generate good discussions and further related experimental work.

**Reasons To Reject:**

In my opinion, the chatbot design methodology, which is based on pure prompting might be messing up the overall experiment. It is not clear whether the work is actually evaluating the veracity of personality self-reporting (which is supposed to be the aim of the study) or the ability of imprinting a personality trait into LLMs and to perform accordingly just based on prompting.

According to the paper one single participant interacted with each chatbot. (It is well known the occurrence of personal biases in this type of annotation tasks, which are commonly alleviated by removing the individual means for each annotator.) It is not clear how reliable are the stats (means and variances) as well as the correlations shown in the tables, as for each task and personality trend only 10 or 20 data points are aggregated.

---

> ### Author Response · Authors · 2025-06-02
>
> We sincerely thank Reviewer ezgD for your detailed and thoughtful feedback. The reviewer raised several important concerns, including the study’s objective and individual evaluation reliability. We address these concerns below and will incorporate valuable suggestions in our revision.
>
> **Study objective clarification**
>
> We appreciate the reviewer’s perspective and the opportunity to clarify our study’s scope. Our primary goal is to examine the validity of the common self-reporting method for evaluating LLM chatbot's personality. In our study, we manipulate LLM chatbot’s personality design by using prompt-based techniques.
>
> Prompt-based personality control is a convenient and the most commonly used method in recent literature [1, 2] for shaping chatbot personality-related behavior. Our prompt design is grounded in prior work and has been validated as effective in eliciting different personality expressions in LLMs [3, 4]. The prompt-based method is indeed effective, especially according to the ‘self-reported’ measure. Such a method is also effective according to the user’s perception, as we observed significant differences in the LLM chatbot’s personality dimensions on different personality settings. By manipulating LLM chatbot’s personality design using such a method, we are allowed to evaluate whether the self-reported personality scores align with human-perceived personality during interaction.
> While self-reporting offers a fast and intuitive method for assessing chatbot traits, our results reveal a mismatch between the self-claimed personality and chatbot's actual behavior in simulated real-world scenarios. This inconsistency raises concerns about the validity of static self-report evaluations.
>
> However, we acknowledge that our study focuses on a single personality-setting method (i.e., prompt-based control). Our findings may not fully generalize to alternative chatbot personality construction (e.g., fine-tuning). We will revise the manuscript to make this limitation and our focus on evaluation validity clearer.
>
>
> **Individual evaluation reliability**
>
> We understand your concern regarding individual bias and the reliability of single-user ratings. To address concerns about individual variability, we aggregate evaluation scores across chatbots sharing the same personality configuration within all task settings (for all high or all low).
>
> Although these aggregated results do not account for finer distinctions between specific tasks, they do consider the effects of high- or low-condition personality settings. With a larger sample size, the observed significance difference between self-report scores and human-perceived personality scores provides more robust evidence for the limitations of relying solely on self-report measures to assess the personality of personalized LLM chatbots. As the results in the table shown below, the confidence intervals indicate that the difference in personality scores assessed by the two methods are meaningfully different from zero. All comparisons are also statistically significant (p < .001) except for the Conscientiousness score under the high setting. These findings suggest a consistent mismatch between self-report and human-perceived scores, further questioning the validity of self-report-based chatbot personality evaluation.
>
> | Trait | High Mean | High CI | High p-value | Low Mean | Low CI | Low p-value |
> |-----:|:---------:|:-------:|:------------:|----------|--------|-------------|
> | Ext   |    -0.8   |  (-1.05, -0.55)  |   <.001***   | 2.19   | (1.93, 2.45) | <.001***    |
> | Agr   |   -0.59   |  (-0.87, -0.30)  |   <.001***   | 0.59     | (0.28, 0.90)  | <.001***    |
> | Con   |   -0.39   |  (-0.67, -0.11)  |    .065**    | 2.31     | (1.98, 2.65)  | <.001***    |
> | Neu   |   -2.12   | (-2.44, -1.79)   |   <.001***   | 0.69     | (0.47, 0.91)  | <.001***    |
> | Ope   |   -1.00   |  (-1.26, -0.74)  |   <.001***   | 2.23     |  (2.01, 2.46) | <.001***    |
>
> Mean differences, 95% confidence intervals (CIs), and p-values between human-perceived and self-reported personality scores under high and low personality settings (computed as human evaluation minus self-report). All comparisons are based on the BFI-2 scale. A p-value < .05 indicates statistical significance at the conventional threshold, while p < .001 denotes very strong evidence that the observed difference is unlikely to be due to chance.
>
> Note: n=50 for each combination of a domain and a personality setting. Ext = Extraversion; Agr = Agreeableness; Con = Conscientiousness; Neu = Neuroticism; Ope = Openness.

---

> ### Author Response · Authors · 2025-06-02
>
> **Statistical significance**
>
> Thank you for your question regarding the statistical significance of our analyses. We have now included all the p-values in our results.
>
> As shown in the following tables, the differences in LLM chatbots’ self-reports between high and low conditions across tasks are both clear and statistically significant. In contrast, while human-perceived scores also reflect differences between conditions, certain traits (e.g., Extraversion and Openness) show weaker or non-significant differences in specific contexts. Overall, the magnitude of change in human-perceived scores is less pronounced than that observed in the self-reports, and the significance levels vary more across tasks. These mismatches between self-report and human-perceived results support the effectiveness of our approach in distinguishing between static metrics and real-world perceptions, and further cast doubt on the validity of self-report methods for personality assessment.
>
> | Trait | Job Interview High | Job Interview Low | Job Interview p | Public Service High | Public Service Low | Public Service p | Social Support High | Social Support Low | Social Support p | Travel Planning High | Travel Planning Low | Travel Planning p | Guided Learning High | Guided Learning Low | Guided Learning p | All High | All Low | All p  |
> |-------|:------------------:|:-----------------:|-----------------|:-------------------:|:------------------:|------------------|:-------------------:|--------------------|------------------|----------------------|---------------------|-------------------|----------------------|---------------------|-------------------|----------|---------|--------|
> | Ext   |        4.86        |        1.15       | < .001***          |         4.63        |        1.04        |       < .001***           |         4.92        | 1.08               |         < .001***         | 4.93                 | 1.04                |        < .001***           | 4.67                 | 1.10                |         < .001***          | 4.80     | 1.08    | < .001*** |
> | Agr   |        3.77        |        1.00       |         < .001***        |         5.00        |        1.03        |       < .001***           |         4.99        | 1.15               |         < .001***         | 4.97                 | 1.00                |      < .001***             | 4.73                 | 1.31                |       < .001***            | 4.70     | 1.10    | < .001*** |
> | Con   |        4.13        |        1.83       |       < .001***          |        -0.04        |        0.20        |    < .001***              |         3.63        | 1.14               |        < .001***          | 5.00                 | 1.01                |       < .001***            | 4.93                 | 1.13                |        < .001***           | 4.54     | 1.23    | < .001*** |
> | Neu   |        4.93        |        1.10       |       < .001***          |         5.00        |        1.00        |     NA             |         4.79        | 1.38               |       < .001***           | 5.00                 | 1.00                |         < .001***          | 4.66                 | 1.01                |          < .001***         | 4.88     | 1.10    | < .001*** |
> | Ope   |        4.60        |        1.03       |        < .001***         |         5.00        |        1.00        |    NA            |         4.30        | 1.13               |         < .001***         | 4.93                 | 1.03                |        < .001***           | 4.93                 | 1.00                |        < .001***           | 4.75     | 1.04    | < .001*** |
>
> Means and p-values comparing self-report scores between high and low conditions, based on the BFI-2 scale. Comparisons within each domain are limited to chatbots explicitly prompted for that domain.  A p-value < .05 indicates statistical significance at the conventional threshold, while p < .001 denotes very strong evidence that the observed difference is unlikely to be due to chance.
>
> Note: NA refers to no variance. Ext = Extraversion; Agr = Agreeableness; Con = Conscientiousness; Neu = Neuroticism; Ope = Openness.

---

> > ### Author Response · Authors · 2025-06-02
> >
> > | Trait | Job Interview High | Job Interview Low | Job Interview p | Public Service High | Public Service Low | Public Service p | Social Support High | Social Support Low | Social Support p | Travel Planning High | Travel Planning Low | Travel Planning p | Guided Learning High | Guided Learning Low | Guided Learning p | All High | All Low |  All p  |
> > |-------|:------------------:|:-----------------:|-----------------|:-------------------:|:------------------:|------------------|:-------------------:|:------------------:|:----------------:|:--------------------:|:-------------------:|:-----------------:|:--------------------:|:-------------------:|:-----------------:|:--------:|:-------:|:-------:|
> > | Ext   |        3.70        |        3.40       | .376           |         4.37        |        2.93        | < .001***           |         3.93        |        3.50        |       .287      |         4.10         |         2.93        |       .012*       |         3.90         |         3.60        |       .404      |   4.00   |   3.27  | < .001*** |
> > | Agr   |        4.20        |        1.57       | < .001***          |         4.41        |        2.07        | .001***            |         4.50        |        1.87        |      < .001***      |         3.27         |         1.50        |       .002**       |         4.10         |         1.43        |       < .001***      |   4.11   |   1.69  | < .001*** |
> > | Con   |        4.00        |        3.60       | .428           |         4.06        |        2.77        | .018*            |         3.93        |        3.97        |       .942      |         4.60         |         3.20        |       .003**       |         4.20         |         1.47        |       .004**       |   4.15   |   3.54  |  .004**  |
> > | Neu   |        2.90        |        1.80       | .009**           |         2.60        |        1.90        | .029*            |         2.27        |        1.70        |       .256      |         3.10         |         1.50        |       .004**       |         2.93         |         2.03        |       < .001***      |   2.76   |   1.78  |  < .001*** |
> > | Ope   |        3.43        |        3.23       | .539           |         3.80        |        3.07        | .048*            |         3.83        |        3.23        |       .098      |         4.00         |         3.27        |       .083       |         3.70         |         3.57        |       .003**       |   3.75   |   3.27  |  .003**  |
> >
> > Means and p-values comparing human-perceived personality scores between high and low conditions, based on the BFI-2 scale. Comparisons within each domain are limited to chatbots explicitly prompted for that domain.  A p-value < .05 indicates statistical significance at the conventional threshold, while p < .001 denotes very strong evidence that the observed difference is unlikely to be due to chance.
> >
> > Note: Ext = Extraversion; Agr = Agreeableness; Con = Conscientiousness; Neu = Neuroticism; Ope = Openness.

---

> > ### Author Response · Authors · 2025-06-02
> >
> > Additionally, we report the results of paired t-tests comparing LLM chatbots’ self-reports and human-perceived scores within the same condition, both under the high condition or the low condition. Most of the p-values are highly significant, further highlighting the differences between self-reported and human-perceived personality scores. This underscores the limitations of relying solely on self-report measures to evaluate the personality of personalized LLM chatbots.
> >
> > | Trait | Job Interview High | Job Interview Low | Public Service High | Public Service Low | Social Support High | Social Support Low | Travel Planning High | Travel Planning Low | Guided Learning High | Guided Learning Low | All High | All Low  |
> > |-------|--------------------|-------------------|---------------------|--------------------|---------------------|--------------------|----------------------|---------------------|----------------------|---------------------|----------|----------|
> > | EXT   | .001***            | <.001***          | .343                | <.001***           | .006**              | <.001***           | .004**               | <.001***            | .041*                | <.001***            | <.001*** | <.001*** |
> > | AGR   | .039*              | .182              | .019*               | .063               | .010***             | .020*              | .002**               | .100                | .013***              | .612                | <.001*** | <.001*** |
> > | CON   | .662               | .003**            | .009**              | .001***            | .409                | <.001***           | .005**               | <.001***            | .036*                | <.001***            | .065**   | <.001*** |
> > | NEU   | <.001***           | .012**            | <.001***            | <.001***           | <.001***            | .318               | .001***              | .009**              | <.001***             | .009**              | <.001*** | <.001*** |
> > | OPE   | <.001***           | <.001***          | <.001***            | <.001***           | .059                | <.001***           | .002*                | <.001***            | .019**               | <.001***            | <.001*** | <.001*** |
> >
> > P-values between self-report and human-perceived personality scores under high or low settings based on the BFI-2 scale. Comparisons within each domain are limited to chatbots explicitly prompted for that domain. A p-value < .05 indicates statistical significance at the conventional threshold, while p < .001 denotes very strong evidence that the observed difference is unlikely to be due to chance.
> >
> > Note: Ext = Extraversion; Agr = Agreeableness; Con = Conscientiousness; Neu = Neuroticism; Ope = Openness.

---

> ### Author Response · Authors · 2025-06-02
>
> **References**
>
> [1] Xintao Wang, Yunze Xiao, Jen-tse Huang, Siyu Yuan, Rui Xu, Haoran Guo, Quan Tu, Yaying Fei, Ziang Leng, Wei Wang, et al. Incharacter: Evaluating personality fidelity in role- playing agents through psychological interviews. In Proceedings of the 62nd Annual Meeting of the Association for Computational Linguistics (Volume 1: Long Papers), pp. 1840–1873, 2024.
>
> [2] Hang Jiang, Xiajie Zhang, Xubo Cao, and Jad Kabbara. Personallm: Investigating the ability of large language models to express big five personality traits. arXiv preprint arXiv:2305.02547, 2023.
>
> [3] Greg Serapio-Garc ́ıa, Mustafa Safdari, Cle ́ment Crepy, Luning Sun, Stephen Fitz, Peter Romero, Marwa Abdulhai, Aleksandra Faust, and Maja Mataric ́. Personality traits in large language models. arXiv preprint arXiv: 2307.00184, 2023.
>
> [4] Juhye Ha, Hyeon Jeon, Daeun Han, Jinwook Seo, and Changhoon Oh. CloChat: Understanding how people customize, interact, and experience personas in large language models. In Proceedings of the 2024 CHI Conference on Human Factors in Computing Systems, pp. 1-24. 2024.

---

> ### Author Response · Authors · 2025-06-09
>
> Dear Reviewer ezgD,
>
> Thank you again for your valuable feedback. We have carefully considered your comments and have done our best to address your concerns regarding the study objective, evaluation reliability, and statistical significance in our response. As we have not yet received any acknowledgment, we wanted to check if there are any remaining points that are unclear or if you have additional concerns. We would be happy to clarify or provide further details before the discussion deadline.

---

> > ### Comment · Reviewer_ezgD · 2025-06-09
> >
> > Thanks for your detailed responses and extra effort to address my concerns and question.

---

> > > ### Author Response · Authors · 2025-06-10
> > >
> > > Dear Reviewer ezgD,
> > >
> > > Thank you again for your kind words. I truly appreciate your thoughtful feedback and the time you took to consider our responses. Your encouragement motivates us to continue improving our work.

---

### Official Review · Reviewer_BH6H · 2025-05-13

**Rating:** 7
**Confidence:** 4
**Ethics Flag:** 1

**Summary:**

This paper examines the validity of using self report methods to measure LLM personality (where LLMs are prompted to respond to various personality scales). The authors build 500 chatbots, with varying personalities and tasks, and have each respond to three validated personality assessments. They then have humans interact with the chatbots and then rate the LLM's personality. Results show that self-reported personality shows low correlation with perceived personality as well as interaction quality, calling into question the validity of the self-report method.

**Questions To Authors:**

* line 36: The Serapio-Garcia reference seems out of place. Are you referencing it with respect to "such as the 70 bipolar adjectives proposed by Goldberg" or "converting personality scale items into open-ended questions"?
* Section 3.1: given your prompt (in Table 8), it seems that each chatbot is prompted to be high or low on a single personality domain, rather than each chatbot having a score on all 5 domains. This should be made explicit.
* Table 1: this is very hard to read, as the font is very small
* Table 1: The caption says n = 500, but that is somewhat misleading as each cell has 10 chatbots (if I understand it correctly).
* Table 1: it's interesting that the neuroticism scores are so low (when compared to the other domains), across all tasks, in both the high and low settings. Is your LLM averse to being neurotic?
* Table 3: You might want to center the coloring at 0, as you have positive correlations colored in red. The reader will look across rows, expecting to see the same color, and drawing conclusions if there are mismatches.

**Reasons To Accept:**

* Using self reports to measure various psychological constructs of LLMs is a popular line of work, and thus assessing the validity of this method is important.
* The use of measurement theory to assess personality is a good choice.
* Despite some issues described below, the set of analyses chosen to evaluate the self-report method are well chosen.

**Reasons To Reject:**

* There is a lot of literature on self vs informatant personality reports (see Kim et al., 2019 for a meta-analysis), none of which is considered in this paper. While informant reports do indeed tend to match self reports, it depends on the relationship between the self and informant. For example, friends/family members are better at rating personality than strangers. Where do LLMs fall on that spectrum? I assume closer to strangers than friends/family members, but also maybe further away (from friends/family) than strangers, given a 10 minute conversation. Similarly, I think we should expect human perceived ratings to correlate less strongly than other self reports (personality scales). So is this a problem with the self-report method or is this just a natural result of the fact that self reports will more strongly correlate with each other than self/informant?

* There are a lot of methodological details that are vague. First, from your prompt it seems like each of the 500 chatbots are prompted with a single (high / low) value for each of the Big 5 domains, which gives 100 chatbots for each domain. Is this correct? If so, then that leaves 400 chatbots which are not prompted on a given domain. So what does this mean for each analysis? Each of your tables lists n = 500, but I don't think that is the case. My read of Table 1 is that each cell has 10 chatbots. (Again, is this correct?) Does this carry over to other tables? For example, in Table 2, in the Extraversion / BFI-2-XS vs BFI-2 cell, is this really a correlation over all 500 chatbots or is this just the Extraversion prompted chatbots (n=100)? Similarly for Table 3, is each cell n = 500? I think Yes here, because I can't imagine how you could compare the Extraversion prompted chatbots to Agreeable prompted chatbots, since they are non-overlapping. This leads into my next question.

* What does the personality distribution look like for the 400 chatbots which aren't prompted for a given domain? It's been shown that unprompted LLMs exhibit little variation in personality (Varadarajan and Giorgi et al., 2025). Also, given positivity/agreeableness biases in LLMs, these 400 chatbots might be high in some domains and low in others. (We get some evidence of this in Table 1, where neuroticism is much lower than all other traits, even when chatbots are prompted to be high.) Taken altogether: given that (1) the majority of the chatbots aren't prompted for a given domain, (2) we don't know what this distribution looks like, and (3) if indeed the sample size is 500 for all analyses, then it's hard to contextualize the results.

* I also wonder, does your prompting method work? While your ANOVA results show that there are significant differences between high and low scores, are these differences meaningful? In many cases both the high and low means are around the same integer value on the 5-point scale. For example, 3.7 vs 3.4 on extraversion for the Job Interview category: is 3.4 actually low extraversion? Again, there is endless personality literature to pull from. What are human ranges for high and low extraversion? Yes, there are some differences between your High and Low groups, but these are mostly with Agreeableness (the other 4 domains look pretty close).

* Two minor issues: First, the Results section often says things like "As shown in Table X, we see some pattern Y and conclude Z", but it's not immediately clear how you are arriving at these patterns/conclusions. This should be spelled out to the reader with references to specific cells, rows/columns, numeric values. Second, are your correlations significant? Are you correcting for multiple comparisons (Bonferroni, for example)?

Kim, H., Di Domenico, S. I., & Connelly, B. S. (2019). Self–other agreement in personality reports: A meta-analytic comparison of self-and informant-report means. Psychological science, 30(1), 129-138.

Varadarajan, V., Giorgi, S., Mangalik, S., Soni, N., Markowitz, D. M., & Schwartz, H. A. (2025, January). The Consistent Lack of Variance of Psychological Factors Expressed by LLMs and Spambots. In Proceedings of the 1stWorkshop on GenAI Content Detection (GenAIDetect) (pp. 111-119).

---

> ### Author Response · Authors · 2025-06-02
>
> We sincerely thank reviewer BH6H for your detailed feedback and for recognizing our contributions. We address all comments below and will revise our manuscript according to your valuable suggestions.
>
> **Informant personality reports**
>
> We sincerely appreciate the reviewer’s attention to this point. Indeed, a substantial body of psychological research has investigated the distinctions between self-report and informant (observer) report methods in personality assessment [1, 2, 3]. Self-reports reflect an individual’s internal view of themselves, while observer-reports provide an external perspective based on how others perceive their behavior. However, the present study is grounded by the motivation that relying solely on chatbots’ self-reported personality scores may be insufficient for evaluating their behavior in real-world interactions. And we want to emphasize that user perception should be the key criterion when evaluating a chatbot’s personality design.
>
> In most practical use cases, users form impressions of chatbots within a limited window of interaction. Our experimental design was intended to simulate this dynamic: participants engaged in brief, task-oriented dialogues with the chatbot to reflect typical real-world human–AI interaction. However, much of the current evaluation of personalized LLM chatbots remains rooted in static chatbots’ self-reports, in which models assess themselves on personality inventories. Our study aims to demonstrate that such evaluations may diverge markedly from the impressions formed by users through direct interaction, which is the end goal for personality design.
>
> As our results indicate, there is often a considerable discrepancy between the LLM chatbots’ self-reported personality and those perceived by human users. This divergence is more related to the mismatch between the self-report format and the interactive, user-facing nature of LLM applications. We argue that human-centered personality assessment which is grounded in actual user experience may be more ecologically valid and reflective of the chatbots’ real-world performance. Our empirical findings provide evidence for these differences and reinforce the value of incorporating user perception as a critical dimension in the evaluation of personalized LLMs.
>
> Moreover, we have explored alternative approaches that offer more nuanced and context-sensitive personality assessments than static self-reporting (see Appendix I). As shown in Table 15, personality scores inferred by the model fine-tuned on human evaluations from human–chatbot conversation transcripts exhibit stronger correlations with human-perceptions than scores obtained through the self-report method. This finding suggests that personality evaluations based on multi-turn interactions provide a more behaviorally grounded assessment than static self-reports, particularly in applied contexts where the alignment between system behavior and user impression is paramount.
>
> **Method** |  **Ext**  |  **Agr**  |  **Con**  |  **Neu**  |  **Ope**  |
> |:---------:|:---------:|:--------:|:--------:|:---------:|-----------|
> |Self-report | 0.20 | 0.59 | 0.25 |  0.33 | 0.21 |
> |Fine-tuned | 0.28 | 0.77 | 0.52 |  0.40 | 0.32 |
>
> Correlation between machine-inferred and human-perceived personality scores (n = 100, test set), compared to correlations between self-reported and human-perceived scores based on the BFI-2 scale.

---

> > ### Comment · Reviewer_BH6H · 2025-06-06
> >
> > First, I would like to thank the authors for the detailed responses! There is a lot to take in and connect back to the manuscript, so I apologize for my delay in responding.
> >
> > **Informant personality reports**
> >
> >     Indeed, a substantial body of psychological research has investigated the distinctions between self-report and informant (observer) report methods in personality assessment [1, 2, 3].
> >
> > Maybe I missed this in the paper, but I think it would be good to discuss this in the paper.
> >
> >     Moreover, we have explored alternative approaches that offer more nuanced and context-sensitive personality assessments than static self-reporting (see Appendix I). As shown in Table 15, personality scores inferred by the model fine-tuned on human evaluations from human–chatbot conversation transcripts exhibit stronger correlations with human-perceptions than scores obtained through the self-report method. This finding suggests that personality evaluations based on multi-turn interactions provide a more behaviorally grounded assessment than static self-reports, particularly in applied contexts where the alignment between system behavior and user impression is paramount.
> >
> > Can you be more specific in how the model in Appendix I was fine tuned? The paper says "human-chatbot conversation samples and their corresponding scores" but which scores are these?
> >
> > I'm not sure I follow your conclusions here: `finding suggests ... provide a more behaviorally grounded assessment`. The results show that human perception correlates higher with machine perception than it does with self reports. How is this "behaviorally grounded"? Where the is behavior here? Isn't it just other (human) perception correlates with other (machine) perception?
> >
> > `However, the present study is grounded by the motivation that relying solely on chatbots’ self-reported personality scores may be insufficient for evaluating their behavior in real-world interactions`. Again, I'm not sure what behavior is being measured here, but also I'm not sure this is how behavioral measures work. One does not observe a person's behavior and then fill out the standard self-report assessment for the person being observed. The behavior itself is quantified (see Jackson et al., 2010).
> >
> > `Our study aims to demonstrate that such evaluations may diverge markedly from the impressions formed by users through direct interaction, which is the end goal for personality design.` I absolutely agree with this, the end goal of personality design may not be the actual personality of the bot but some outcome from the chat interaction (e.g., UEQ). But here you are saying perceptions ("impressions") are the thing we care about, rather than the chatbot's behavior. Your results show that the perception of personality better aligns with other perceptions (UEQ) than self-reports do. Another way to frame this is that 2nd person ratings correlate more strongly with themselves than 1st vs 2nd person ratings. So I go back to my question in my original review: is this a problem with the self-report method or is this just a natural result of the fact that (1st person) self reports will more strongly correlate with each other than self/informant (1st vs 2nd)? (The same with 2nd person measures, such as perceived personality and UEQ: they will correlate with each other at high levels than 1st vs 2nd.) I agree with a lot of what you say (`...the value of incorporating user perception as a critical dimension in the evaluation of personalized LLMs.`, `...user perception should be the key criterion when evaluating a chatbot`, etc) but I don't think that's the same as saying "self reports are not valid" (or "2nd person evaluations are behavioral") when it could just be "1st and 2nd person assessments measure different things" (and there is a lot of evidence to support this claim).
> >
> > Jackson, J. J., Wood, D., Bogg, T., Walton, K. E., Harms, P. D., & Roberts, B. W. (2010). What do conscientious people do? Development and validation of the Behavioral Indicators of Conscientiousness (BIC). Journal of research in personality, 44(4), 501-511.
> >
> > **Statistical significance and correction**
> >
> >     Actually, our study does not involve multiple comparisons
> >
> > Can you please further justify this statement? I agree that the tasks and the domains are different for each test, but the hypothesis you are testing is the same across tasks/domains. In the end you are pooling the results across all of the tests to make a single inference (i.e., you are using multiple tasks to strengthen your claims, as in "look! this pattern is task/domain independent").
> >
> > Re the statement `Therefore, we were not making repeated comparisons on the same dataset`: I don't think this is a fair assessment. The paper conceptualizes the dataset as a single dataset (`n=500`; where you would need FDR), rather than a Study 1, Study 2, Study 3, etc. design (where you wouldn't).

---

> > > ### Author Response · Authors · 2025-06-06
> > >
> > > **Overall Reply to Reviewer BH6H**
> > >
> > > We sincerely thank you for your valuable suggestions. Your feedback has been highly constructive, providing important insights and guidance for our revisions and helping us further improve the quality of our paper. We are also truly grateful to the COLM conference for providing such a platform, allowing us to receive in-depth comments and support from many outstanding reviewers.
> > >
> > > **Reply to ‘Informant personality reports’**
> > >
> > > Thank you again for highlighting the importance of including a discussion on informant personality reports. We will expand the motivation in the introduction section to include this part, highlighting theoretical and empirical findings from the personality literature regarding self-report and informant-report methods. This will help illustrate how these two approaches differ in terms of information sources, types of bias (e.g., self-bias versus observer bias), and how they may complement each other across different personality dimensions. We will then connect this to the aim of our paper: comparing LLM chatbot self-reports with user-perceived reports to demonstrate the importance of considering user perception as a key chatbot evaluation criterion, emphasizing how this differs from relying solely on self-reports.
> > >
> > > **Reply to ‘how the model was fine tuned?’**
> > >
> > > Thank you for your question regarding the fine-tuning details. In the experiment, we split all the transcripts of human-chatbot conversions and their corresponding human evaluations of chatbot personality into training and testing sets using an 80:20 ratio. Human perceived chatbot personality scores were collected using the BFI-XS scale, which balances efficiency with robust predictive validity.  With both the transcript and human-perceived personality scores as labels, we fine-tuned GPT-4o on the training set using a prompt that included task-specific instructions (shown in Table 14), and evaluated the model on the testing set to assess the alignment between its output and actual human perceptions. The fine-tuning was conducted via the API from OpenAI.
> > >
> > > **Reply to ‘how behaviorally grounded’**
> > >
> > > In our study, "behavior" does not refer to directly observed physical actions involving body language or specific decisions, but rather to the outputs generated by an LLM chatbot during a task. These behavioral expressions include response style, intent, problem-solving approaches, and tone of language. These outputs are received and perceived by human participants in our study to make personality judgments based on these interactions. Note that in our study, the human participants interact with the chatbot to complete the task rather than being given a transcript to rate. In other words, we treat the LLM outputs exhibited during the task as observable “behavioral expressions.” This aligns with how most users interacting with LLM chatbots intuitively experience their behavior and how they form personality assessments (e.g., users tend to interpret them in human-like terms, treating the chatbot’s responses as expressions of underlying personality traits [1].). For instance, if the chatbot uses enthusiastic tones or emotionally charged language, users are likely to perceive it as extraverted, even if its self-report doesn't reflect that trait.
> > >
> > > This approach is consistent with the Realistic Accuracy Model (RAM) [2], which emphasizes inferring personality traits based on perceived behavioral cues from others. Although we have not explicitly provided language behavior anchors in this study, this remains a promising direction for future work. The aggregation of these user perceptions can form a relatively stable “group consensus” or commonsense judgment, reflecting an overall assessment of the LLM’s language behavior.
> > >
> > > [1] Nass, C., & Moon, Y. (2000). Machines and mindlessness: Social responses to computers. Journal of social issues, 56(1), 81-103.
> > >
> > > [2] Funder, D. C. (1995). On the accuracy of personality judgment: a realistic approach. Psychological review, 102(4), 652.

---

> > > > ### Author Response · Authors · 2025-06-06
> > > >
> > > > **Reply to ‘1st vs 2nd person ratings…’**
> > > >
> > > > We absolutely agree with your insightful comments, especially the point that first-person (LLM Chatbot self-report) and second-person (user-perceived) assessments may be measuring different constructs rather than one being more “valid” than the other. This distinction was not clearly articulated in our original draft, and we will revise the camera-ready version to reflect this more accurately.
> > > >
> > > > Our main argument is not that LLM chatbot self-reports are invalid per se, but that they may not fully capture the behavioral cues users actually respond to during real interactions. In our framing, we treat the chatbot’s behavior (e.g., [1]) during task completion as the observable expression of personality, which is what users primarily base their perceptions on. For instance, if the chatbot uses enthusiastic tones or emotionally charged language, users are likely to perceive it as extraverted, even if its self-report doesn't reflect that trait. We argue that such behaviours are what the LLM self-reports should capture, given that this method has been widely adopted in the field to measure LLMs’ “personality”. Then the validity of this method should be grounded on the LLM chatbot’s behavioral manifestation. In our study, we capture such manifestations through our study participants' perceptions.
> > > >
> > > > Thus, we will adjust our language to clarify that:
> > > > Self-reports from LLMs may reflect the model’s learned associations between prompts and trait descriptors (scale items), rather than a grounded behavioral manifestation of personality (e.g., [2] LLMs may learn associations through prompts rather than truly understanding concepts).
> > > >
> > > > In contrast, user-perceived personality reflects how users interpret the chatbot’s actual behavior in interactional contexts, which often aligns more closely with downstream outcomes (e.g., UEQ ratings).
> > > >
> > > > We will tone down our claims that self-report is “invalid,” but rather limited in capturing user-facing behavioral outcomes. Highlight the importance of aligning personality evaluation methods with intended real-world use cases, where user impression and interaction experience matter most.
> > > >
> > > > [1] Schwartz, H. A., Eichstaedt, J. C., Kern, M. L., Dziurzynski, L., Ramones, S. M., Agrawal, M., ... & Ungar, L. H. (2013). Personality, gender, and age in the language of social media: The open-vocabulary approach. PloS one, 8(9), e73791.
> > > >
> > > > [2] Wei, J., Tay, Y., Bommasani, R., Raffel, C., Zoph, B., Borgeaud, S., Yogatama, D., Bosma, M., Zhou, D., Metzler, D. and Chi, E.H., (2022). Emergent abilities of large language models. arXiv preprint arXiv:2206.07682.

---

> > > > > ### Author Response · Authors · 2025-06-06
> > > > >
> > > > > **Reply to ‘Statistical significance and correction’**
> > > > >
> > > > > Thank you very much for pointing this out. This is indeed a lack of rigor in our previous response. What you mentioned here is absolutely correct, the goal of the analysis is not to independently verify the effect of each individual task, but rather to strengthen the overall conclusion through multiple tasks. Therefore, significance correction should indeed be applied. We will add further analysis and explanation on this point in our next version.
> > > > >
> > > > > With the application of significance correction, our overall conclusions remain unaffected, as demonstrated in our previous response. As we noted previously, most of our significant p-values are below .001. This threshold is more stringent than the adjusted alpha level that would result from correcting for, say, 50 comparisons using either Bonferroni correction (α = .001) or FDR procedures such as Benjamini-Hochberg. Thus, even under false discovery rate control (e.g., FDR q < 0.05), our main effects would still be retained as statistically significant. In this context, applying FWER/FDR correction would not alter the statistical interpretation of our key results, and would not undermine the broader theoretical conclusions drawn from them.
> > > > >
> > > > > | Trait | Job Interview High | Job Interview Low | Public Service High | Public Service Low | Social Support High | Social Support Low | Travel Planning High | Travel Planning Low | Guided Learning High | Guided Learning Low |
> > > > > |-------|--------------------|-------------------|---------------------|--------------------|---------------------|--------------------|----------------------|---------------------|----------------------|---------------------|
> > > > > | EXT   | .002**               | <.001***             | .365                | <.001***              | .011*                | <.001***              | .007**                | .001***               | .050*                | <.001***               |
> > > > > | AGR   | .049*              | .202             | .027*               | .075*              | .015*               | .028*              | .004**                | .117               | .020*                | .625               |
> > > > > | CON   | .662              | .005**             | .014*               | .002**              | .426               | <.001***              | .009**                | <.001***               | .046*                | <.001***               |
> > > > > | NEU   | <.001***              | .018*             | <.001***               | <.001***              | <.001***               | .346              | .003**                | .014*               | .001***                | .014*               |
> > > > > | OPE   | .005**              | <.001***             | <.001***               | <.001***              | .132               | <.001***              | .028*                | <.001***               | .011*                | <.001***               |
> > > > >
> > > > > FDR-adjusted p-values, referred to as q-values, were calculated using the Benjamini-Hochberg procedure to compare self-reported and human-perceived personality scores under high and low settings based on the BFI-2 scale. Comparisons within each domain are limited to chatbots explicitly prompted for that domain. A p-value < .05 indicates statistical significance at the conventional threshold, while p < .001 denotes very strong evidence that the observed difference is unlikely to be due to chance.
> > > > > Note: Ext = Extraversion; Agr = Agreeableness; Con = Conscientiousness; Neu = Neuroticism; Ope = Openness.

---

> > > ### Author Response · Authors · 2025-06-09
> > >
> > > Dear Reviewer BH6H,
> > >
> > > Thank you for your thoughtful feedback and helpful suggestions. We have carefully revised our response to include fine-tuning details, explanations on the behavioral grounding of our evaluation and the distinction between first- and second-person personality assessments, and analysis on statistical significance with correction. If any part of our response remains unclear or if there are additional points you would like us to elaborate on, we would appreciate the opportunity to clarify further. We’d be happy to address any remaining concerns ahead of the discussion deadline.

---

> > > > ### Comment · Reviewer_BH6H · 2025-06-10
> > > >
> > > > Thank you for all of your responses! In case it's not clear, there is a lot to like about this paper. I enjoyed reading it and our exchange here.
> > > >
> > > > **Reply to ‘how behaviorally grounded’**
> > > >
> > > > Yes, I realize that by "behavior" you mean "the interactions and language from the conversation". My questions were more "what is the quantified behavior" (sorry for any confusion).
> > > >
> > > > I haven't read the Funder paper, do they really measure personality in the same way you do (an informant observes a behavior and then fills out a standard personality battery about the other person)? I've never seen this (and everything I've seen is like the Jackson paper I listed, where the actual behavior is quantified), but also I'm not an expert.
> > > >
> > > > One *could* look at the behavior, for example using the methods in the Schwartz paper you cited. While this would be super interesting, this is outside of the scope of the work. I think without this piece (explicitly quantifying behavior), then you can't argue "behaviorally grounded". But I don't think you really need this.
> > > >
> > > > **Reply to ‘Statistical significance and correction’**
> > > >
> > > > Agreed that many of your results remain significant (though some will not, as in the `Means and p-values comparing human-perceived personality scores between high and low conditions, based on the BFI-2 scale` table). But, at least for me, knowing the exact sample sizes plus these tests gives me more confidence in the results.
> > > >
> > > > **Reply to ‘1st vs 2nd person ratings…’**
> > > >
> > > > This is my personal read of the results: in LLM / human interactions (as opposed to human / human interactions), we mostly care about 2nd person perceptions. If LLM self-reports don't predict these things then they are useless (in some sense). Does this mean LLM self-reports (1st person) are invalid? I don't think you've shown this and some of your results show that they are indeed valid (self-reports correlate with the prompts).
> > > >
> > > > **Summary**
> > > >
> > > > There is no need to reply to the above (unless you'd really like to). You've addressed most of my concerns. While I do feel strongly about the above points (i.e., I think your framing is not correct), I also realize I am the minority reviewer. If the authors, other reviewers, or AC feel strongly in another direction then I am fine to cede my point. Thank you again for the exchange!

---

> > > > > ### Author Response · Authors · 2025-06-10
> > > > >
> > > > > Dear Reviewer BH6H,
> > > > >
> > > > > Thank you for your valuable feedback and for recognizing our contributions. We've truly enjoyed our exchange with you. We are pleased to have addressed some of your concerns, we are wondering if you can adjust your rating to reflect that. Regardless, we are very appreciative of your last comment. In fact, the framing of the entire paper has been greatly revised based on your comments.
> > > > >
> > > > > **Reply to ‘How behaviorally grounded’:**
> > > > >
> > > > > We believe there may be two approaches to evaluating behavior. The first, as you mentioned, involves the explicit quantification of behavior, which we refer to here as “behavior frequency”. The second approach is like Funder’s RAM model, which relies on “behavior impressions”, that is, observers (informants) form impressions based on observed behaviors. In this method, behavior serves as the source of information, but what is quantified is the observer’s report, not the behavior itself. We will address “behavior frequency” as an alternative when we are discussing the “behaviorally grounded” approach.
> > > > >
> > > > > **Reply to ‘Statistical significance and correction’:**
> > > > >
> > > > > Thank you very much. We will ensure that our final version clearly specifies the sample size and corrections. We sincerely appreciate your valuable suggestions, which have played an important role in guiding our revisions to the corresponding sections of the manuscript.
> > > > >
> > > > > **Reply to ‘1st vs 2nd person rating’:**
> > > > >
> > > > > Thank you very much for your insightful comments. As you rightly pointed out, our analysis is not intended to dismiss the role of LLM self-reports (1st person), but rather to examine the utility of LLM self-reports, especially in real interactions between LLM-based chatbots and humans.
> > > > >
> > > > > One way to think about this is to consider self-reports as a form of “exam response”: when we use personality inventories (e.g., psychometric tests based on the Big Five) to “test” an LLM, we are essentially giving it a well-defined task with a constrained set of expected answers. LLMs excel at such tasks. They analyze the semantics of each item and respond accordingly, often drawing from the persona information provided in the prompt. In contrast, real interaction is more like “improvised performance”: it unfolds in a dynamic, open-ended, and unpredictable context. Each user question, tone, or follow-up shapes the trajectory of the conversation. In these situations, it is much harder for the LLM to consistently align its actual language behavior with the persona information provided earlier.
> > > > >
> > > > > While the validity of personality inventories is based on empirical behavioral evidence from humans, showing relatively consistent human responses on those personality scales and their actual behaviors, such assumption has not been tested for LLMs. However, much of the current work assumes the validity of using personality inventories for LLMs. Our work challenges this assumption by demonstrating that LLMs may struggle to maintain alignment between self-reports and actual linguistic behavior in interactive and task-oriented settings.

---

> > > > > > ### Comment · Reviewer_BH6H · 2025-06-10
> > > > > >
> > > > > > I've updated my score from a 4 to a 7, under the assumption that the above will make it into the next version.

---

> > > > > > > ### Author Response · Authors · 2025-06-10
> > > > > > >
> > > > > > > Dear Reviewer BH6H,
> > > > > > >
> > > > > > > Thank you very much for taking the time to revisit our response and update your score. We sincerely appreciate your thoughtful engagement and constructive feedback. We are fully committed to incorporating those improvements in the revised version, and your evaluation is highly valued.

---

> ### Author Response · Authors · 2025-06-02
>
> **Methodology details clarification**
>
> Thank you for raising questions about the interpretation of sample sizes across analyses. To clarify, each chatbot is prompted with a specific personality instruction for one of the five Big Five traits at either high or low level, and is assigned a context setting. This results in 500 different chatbot personality descriptions in total (i.e., 10 chatbots × 2 levels × 5 tasks × 5 domains). The n = 500 noted in each table caption refers to the total number of chatbots evaluated. For a given domain, 100 chatbots are prompted for that trait, while the remaining 400 are not.
>
> In Table 1, each cell represents 10 chatbots with a specific combination of personality domain, level, and task. This setup accounts for the full set of 500 chatbots.
>
> Table 2 presents correlation analyses across all 500 chatbots, using one-to-one comparisons between each chatbot’s self-reported and human-perceived scores within a given personality domain. Although only 100 chatbots are directly prompted for a given domain, we include all 500 in the correlation analysis based on the assumption that a valid automatic evaluation method should align with human evaluations across all domains, regardless of whether the trait was explicitly prompted. Thus, each cell in Table 2 is based on 500 data points.
>
> Table 3 reports correlations between different personality traits within each chatbot to evaluate discriminant validity. As each chatbot has evaluations scores for all five traits, this analysis is also conducted across all 500 chatbots, and each cell reflects 500 samples.

---

> ### Author Response · Authors · 2025-06-02
>
> **Validity of the personality-setting method**
>
> We appreciate your concerns about the effectiveness of our personality-setting method. We measure its effectiveness through two perspectives. One approach, widely used in both research and practical application, involves examining the differences in self-reports of LLM chatbots under high and low conditions for each personality domain. However, we argue that relying on this approach alone is insufficient. We also assess the differences in human-perceived personality between the high and low conditions.
>
> The following tables present the results from both approaches. As shown, the differences in LLM chatbots’ self-reports between high and low conditions across tasks are clear and statistically significant. In contrast, while human-perceived scores also reflect differences between conditions, some traits (e.g., Extraversion and Openness) show weaker or non-significant differences in certain contexts. The overall magnitude of its change is less extreme than in the self-report method, and the significance varies more across tasks. The mismatches with self-report results verifies the effectiveness of our method in distinguishing the static metric and the real-world perception, and further doubts the validity of self-report methods for personality evaluation.
>
> | Trait | Job Interview High | Job Interview Low | Job Interview p | Public Service High | Public Service Low | Public Service p | Social Support High | Social Support Low | Social Support p | Travel Planning High | Travel Planning Low | Travel Planning p | Guided Learning High | Guided Learning Low | Guided Learning p | All High | All Low | All p  |
> |-------|:------------------:|:-----------------:|-----------------|:-------------------:|:------------------:|------------------|:-------------------:|--------------------|------------------|----------------------|---------------------|-------------------|----------------------|---------------------|-------------------|----------|---------|--------|
> | Ext   |        4.86        |        1.15       | < .001***          |         4.63        |        1.04        |       < .001***           |         4.92        | 1.08               |         < .001***         | 4.93                 | 1.04                |        < .001***           | 4.67                 | 1.10                |         < .001***          | 4.80     | 1.08    | < .001*** |
> | Agr   |        3.77        |        1.00       |         < .001***        |         5.00        |        1.03        |       < .001***           |         4.99        | 1.15               |         < .001***         | 4.97                 | 1.00                |      < .001***             | 4.73                 | 1.31                |       < .001***            | 4.70     | 1.10    | < .001*** |
> | Con   |        4.13        |        1.83       |       < .001***          |        -0.04        |        0.20        |    < .001***              |         3.63        | 1.14               |        < .001***          | 5.00                 | 1.01                |       < .001***            | 4.93                 | 1.13                |        < .001***           | 4.54     | 1.23    | < .001*** |
> | Neu   |        4.93        |        1.10       |       < .001***          |         5.00        |        1.00        |     NA             |         4.79        | 1.38               |       < .001***           | 5.00                 | 1.00                |         < .001***          | 4.66                 | 1.01                |          < .001***         | 4.88     | 1.10    | < .001*** |
> | Ope   |        4.60        |        1.03       |        < .001***         |         5.00        |        1.00        |    NA            |         4.30        | 1.13               |         < .001***         | 4.93                 | 1.03                |        < .001***           | 4.93                 | 1.00                |        < .001***           | 4.75     | 1.04    | < .001*** |
>
> Means and p-values comparing self-report scores between high and low conditions, based on the BFI-2 scale. Comparisons within each domain are limited to chatbots explicitly prompted for that domain.  A p-value < .05 indicates statistical significance at the conventional threshold, while p < .001 denotes very strong evidence that the observed difference is unlikely to be due to chance.
>
> Note: NA refers to no variance. Ext = Extraversion; Agr = Agreeableness; Con = Conscientiousness; Neu = Neuroticism; Ope = Openness.

---

> > ### Author Response · Authors · 2025-06-02
> >
> > | Trait | Job Interview High | Job Interview Low | Job Interview p | Public Service High | Public Service Low | Public Service p | Social Support High | Social Support Low | Social Support p | Travel Planning High | Travel Planning Low | Travel Planning p | Guided Learning High | Guided Learning Low | Guided Learning p | All High | All Low |  All p  |
> > |-------|:------------------:|:-----------------:|-----------------|:-------------------:|:------------------:|------------------|:-------------------:|:------------------:|:----------------:|:--------------------:|:-------------------:|:-----------------:|:--------------------:|:-------------------:|:-----------------:|:--------:|:-------:|:-------:|
> > | Ext   |        3.70        |        3.40       | .376           |         4.37        |        2.93        | < .001***           |         3.93        |        3.50        |       .287      |         4.10         |         2.93        |       .012*       |         3.90         |         3.60        |       .404      |   4.00   |   3.27  | < .001*** |
> > | Agr   |        4.20        |        1.57       | < .001***          |         4.41        |        2.07        | .001***            |         4.50        |        1.87        |      < .001***      |         3.27         |         1.50        |       .002**       |         4.10         |         1.43        |       < .001***      |   4.11   |   1.69  | < .001*** |
> > | Con   |        4.00        |        3.60       | .428           |         4.06        |        2.77        | .018*            |         3.93        |        3.97        |       .942      |         4.60         |         3.20        |       .003**       |         4.20         |         1.47        |       .004**       |   4.15   |   3.54  |  .004**  |
> > | Neu   |        2.90        |        1.80       | .009**           |         2.60        |        1.90        | .029*            |         2.27        |        1.70        |       .256      |         3.10         |         1.50        |       .004**       |         2.93         |         2.03        |       < .001***      |   2.76   |   1.78  |  < .001*** |
> > | Ope   |        3.43        |        3.23       | .539           |         3.80        |        3.07        | .048*            |         3.83        |        3.23        |       .098      |         4.00         |         3.27        |       .083       |         3.70         |         3.57        |       .003**       |   3.75   |   3.27  |  .003**  |
> >
> > Means and p-values comparing human-perceived personality scores between high and low conditions, based on the BFI-2 scale. Comparisons within each domain are limited to chatbots explicitly prompted for that domain.  A p-value < .05 indicates statistical significance at the conventional threshold, while p < .001 denotes very strong evidence that the observed difference is unlikely to be due to chance.
> >
> > Note: Ext = Extraversion; Agr = Agreeableness; Con = Conscientiousness; Neu = Neuroticism; Ope = Openness.

---

> > ### Author Response · Authors · 2025-06-02
> >
> > Additionally, we calculated p-values between self-report and human-perceived personality scores under high or low settings based on the BFI-2 scale. Strong statistical significance (p < .001) is observed across most domains and tasks, suggesting consistent patterns of divergence between self-report and human-evaluated scores, and effectiveness of our prompt design.
> >
> > | Trait | Job Interview High | Job Interview Low | Public Service High | Public Service Low | Social Support High | Social Support Low | Travel Planning High | Travel Planning Low | Guided Learning High | Guided Learning Low | All High | All Low  |
> > |-------|--------------------|-------------------|---------------------|--------------------|---------------------|--------------------|----------------------|---------------------|----------------------|---------------------|----------|----------|
> > | EXT   | .001***            | <.001***          | .343                | <.001***           | .006**              | <.001***           | .004**               | <.001***            | .041*                | <.001***            | <.001*** | <.001*** |
> > | AGR   | .039*              | .182              | .019*               | .063               | .010***             | .020*              | .002**               | .100                | .013***              | .612                | <.001*** | <.001*** |
> > | CON   | .662               | .003**            | .009**              | .001***            | .409                | <.001***           | .005**               | <.001***            | .036*                | <.001***            | .065**   | <.001*** |
> > | NEU   | <.001***           | .012**            | <.001***            | <.001***           | <.001***            | .318               | .001***              | .009**              | <.001***             | .009**              | <.001*** | <.001*** |
> > | OPE   | <.001***           | <.001***          | <.001***            | <.001***           | .059                | <.001***           | .002*                | <.001***            | .019**               | <.001***            | <.001*** | <.001*** |
> >
> > P-values between self-report and human-perceived personality scores under high or low settings based on the BFI-2 scale. Comparisons within each domain are limited to chatbots explicitly prompted for that domain. A p-value < .05 indicates statistical significance at the conventional threshold, while p < .001 denotes very strong evidence that the observed difference is unlikely to be due to chance.
> >
> > Note: Ext = Extraversion; Agr = Agreeableness; Con = Conscientiousness; Neu = Neuroticism; Ope = Openness.

---

> ### Author Response · Authors · 2025-06-02
>
> **Statistical significance and correction**
>
> Thank you for your question regarding the statistical significance of our analyses. We have added p-value calculations. Actually, our study does not involve multiple comparisons. The comparisons we conducted were all focused on specific domains within specific tasks. For example, we examined the correlation between chatbots' self-reported and human-perceived extraversion in the Job Interview task under conditions of high or low extraversion. Therefore, we were not making repeated comparisons on the same dataset, which helps avoid the cumulative increase in the probability of Type I Error.
>
> In our study, most of the significant p-values are below .001. This level is not only far below the conventional α = .05 threshold, but also indicates that these results would still be statistically significant even if a large number of comparisons were made. For instance, if α = .05 were corrected for 50 comparisons, the threshold would be .001. Therefore, we believe that our core findings are highly robust and that their significance is not a product of chance due to multiple comparisons.
>
> Regarding the relatively larger differences observed in Agreeableness compared to other domains, one possible explanation is that some traits may be more easily conveyed through language and interaction style than others [4]. This may limit human perception of those traits. We will include this point in the Limitations section to acknowledge differences in trait observability.
>
> **Result interpretation clarification**
>
> Thank you for this helpful suggestion. In the revised version, we will improve the clarity of our analysis by explicitly referencing specific data points when describing observed patterns and conclusions in the Results section.
>
> **Citation clarification**
>
> Thank you for the question regarding the cited reference. To clarify, the Serapio-Garcia reference is intended to support the mapping of Goldberg’s 70 bipolar adjectives to the Big Five domains and their lower-order personality facets. We will revise the sentence to distinguish between Goldberg’s original contribution and Serapio-Garcia’s mapping work to avoid confusion.
>
> **Prompt design clarification**
>
> Thank you for the suggestion to further clarify our prompt design. As noted, each chatbot is prompted to exhibit either a high or low level on one single personality domain, with the other domains left unconstrained. We will revise Section 3.1 to make chatbot personality design more explicit.
>
>
> **Lower Neuroticism scores**
>
> In response to your question about the relatively low human-perceived Neuroticism scores across both high and low conditions in different tasks, this is indeed a very interesting finding. When prompting LLM chatbots to self-report, the difference between the high and low conditions is actually quite pronounced—scores around 5 for the high condition and around 1 for the low condition, with statistically significant p-values. However, this distinction is not reflected in the human-perceived ratings, which remain relatively low across both settings.
>
> One possible explanation is that, due to safety alignment and RLHF during post-training, LLMs may implicitly suppress negative or anxious traits during actual task performance, even when explicitly prompted to adopt a neurotic persona. This aligns with recent findings that LLMs tend to default to more positive or neutral behaviors [5, 6]. Another possibility is that certain tasks inherently make it difficult for users to perceive high neuroticism, especially if such traits conflict with the intended function or design goals of most LLM chatbots. We will include this interpretation in the discussion section and highlight it as a potential limitation of prompt-based personality design.
>
> Overall, our results point to the limitations of relying solely on self-report measures to assess the personality of personalized LLM chatbots.
>
> **Table format**
>
> Thank you for your helpful suggestions on table formatting. For Table 1, we will increase the font size and adjust the layout to improve clarity. In the caption, n = 500 indicates the total number of chatbots evaluated. Each cell represents 10 chatbots corresponding to a specific combination of personality domain, level, and task. Thus, 10 chatbots × 2 levels × 5 tasks × 5 domains = 500 chatbots in total. We will revise the caption to avoid confusion.
>
> For Table 3, we agree that centering the color scale at zero would enhance visual clarity. We will apply a consistent, zero-centered color scale across rows to better highlight the direction and magnitude of the correlations, reducing the risk of misinterpretation.

---

> ### Author Response · Authors · 2025-06-02
>
> **References**
>
> [1] Robert R. McCrae, and Paul T. Costa. Validation of the five-factor model of personality across instruments and observers. Journal of Personality and Social Psychology, 52(1), pp. 81–90.1987.
>
> [2] James J. Connolly, Erin J. Kavanagh, and Chockalingam Viswesvaran. The Convergent Validity between Self and Observer Ratings of Personality: A meta-analytic review. International Journal of Selection and Assessment, 15(1), pp. 110–117. 2007.
> [3] Brian S. Connelly, and Deniz S. Ones. An other perspective on personality: Meta-analytic integration of observers' accuracy and predictive validity. Psychological Bulletin, 136(6), pp. 1092–1122. 2010.
>
> [4] François Mairesse, and Marilyn A. Walker. Towards personality-based user adaptation: psychologically informed stylistic language generation. User Modeling and User-Adapted Interaction, pp. 227-278. 2010.
>
> [5] Aleksandra Sorokovikova, Natalia Fedorova, A. I. Toloka, Sharwin Rezagholi, Technikum Wien, and Ivan P. Yamshchikov. LLMs Simulate Big Five Personality Traits: Further Evidence. In The 1st Workshop on Personalization of Generative AI Systems, pp. 83-87. 2024.
>
> [6] Seungbeen Lee, Seungwon Lim, Seungju Han, Giyeong Oh, Hyungjoo Chae, Jiwan Chung, Minju Kim et al. Do llms have distinct and consistent personality? trait: Personality testset designed for llms with psychometrics. arXiv preprint arXiv:2406.14703. 2024.

---

### Decision · Program_Chairs · 2025-07-08

**Decision:**

Accept

**Comment:**

This paper investigates the validity of using self-report questionnaires (personality scales) to measure personality traits in LLM-powered chatbots. The authors create 500 chatbots with varied personalities (via prompting) and tasks, administer three established personality assessments to the chatbots ("self-report"), and have human participants interact with and rate the chatbots' personalities ("perceived personality"). The core finding is a low correlation between chatbot self-reported personality, human-perceived personality, and interaction quality. This challenges the common practice of using self-reports as a reliable measure of LLM personality.

Pros:
1. The study employs a large-scale design (500 chatbots, multiple personality scales, human evaluations) across diverse tasks, providing substantial data for analysis.
2. Leveraging validated psychometric personality assessments (e.g., BFI-2-XS) strengthens the foundation of the measurement approach.
3. The application of measurement theory and statistical analyses (e.g., ANOVA, correlations) to evaluate validity is generally appropriate and well-chosen.


Cons:
1. The core criticism is that the study potentially confounds manipulating personality via prompting with measuring it via self-report. It's unclear if the low validity stems from inherent limitations of self-reports for LLMs or from the specific prompting method failing to instill stable, discernible personalities (e.g., small differences between "High" and "Low" groups in Table 1).
2. Relying on only one human rater per chatbot introduces significant noise and potential bias, undermining the reliability of the perceived personality and interaction quality measures. Small per-condition sample sizes (e.g., n=10 or 20) further weaken statistical power and generalizability.
3. While ANOVA shows statistically significant differences between High/Low groups for prompted traits, the practical meaningfulness is doubtful. Mean scores for "High" and "Low" groups often fall very close on the scale (e.g., 3.7 vs. 3.4 for Extraversion in Job Interview), suggesting the prompting manipulation may not have created distinctly high or low personality expressions as intended.

[Automatically added comment] At least one review was discounted during the decision process due to quality]